# RETHINKING POLICY DIVERSITY IN ENSEMBLE POLICY GRADIENT IN LARGE-SCALE REINFORCEMENT LEARNING

**Naoki Shitanda**[1,2]**, Motoki Omura**[1]**, Tatsuya Harada**[1,2]**, Takayuki Osa**[2]
`{shitanda,omura,harada}@mi.t.u-tokyo.ac.jp`
`takayuki.osa@riken.jp`

[1]The University of Tokyo, Tokyo, Japan
[2]RIKEN Center for Advanced Intelligence Project, Tokyo, Japan

## ABSTRACT

Scaling reinforcement learning to tens of thousands of parallel environments requires overcoming the limited exploration capacity of a single policy. Ensemble-based policy gradient methods, which employ multiple policies to collect diverse samples, have recently been proposed to promote exploration. However, merely broadening the exploration space does not always enhance learning capability, since excessive exploration can reduce exploration quality or compromise training stability. In this work, we theoretically analyze the impact of inter-policy diversity on learning efficiency in policy ensembles, and propose Coupled Policy Optimization which regulates diversity through KL constraints between policies. The proposed method enables effective exploration and outperforms strong baselines such as SAPG, PBT, and PPO across multiple tasks, including challenging dexterous manipulation, in terms of both sample efficiency and final performance. Furthermore, analysis of policy diversity and effective sample size during training reveals that follower policies naturally distribute around the leader, demonstrating the emergence of structured and efficient exploratory behavior. Our results indicate that diverse exploration under appropriate regulation is key to achieving stable and sample-efficient learning in ensemble policy gradient methods. Project page at `https://naoki04.github.io/paper-cpo/`.

## 1 INTRODUCTION

With the advent of GPU-based massively parallel physics simulators such as Isaac Gym (Makoviychuk et al., 2021) and Genesis (Genesis Authors, 2024), it has become feasible to collect data from over tens of thousands of environments simultaneously for robot deep reinforcement learning (RL). Given the inherently trial-and-error nature of RL, such parallelism has the potential to dramatically improve learning efficiency for high-dimensional and complex tasks, such as dexterous hand manipulation. However, recent work (Singla et al., 2024) has reported that simply increasing the amount of data does not necessarily lead to improved learning efficiency in on-policy methods like PPO (Schulman et al., 2017). This result suggests that simply using a single policy in massively parallelized environments does not sufficiently diversify exploration and thus cannot significantly improve learning efficiency.

To address these challenges, agent ensemble approaches have been proposed to collect diverse samples. Recent work (Singla et al., 2024) introduced a leader-follower framework shown in Fig. 1(a), in which one leader agent and multiple followers are each assigned to separate blocks of parallel environments. Each follower performs independent on-policy learning, while the leader aggregates off-policy samples from followers using importance sampling (IS). Unlike other agent ensemble methods (Aleksei Petrenko, 2023; Li et al., 2023a), this enables the use of all collected data without discarding any samples, thereby facilitating diverse exploration. Their approach has demonstrated significantly improved learning performance over non-aggregating methods like DexPBT (Aleksei Petrenko, 2023), as well as over off-policy methods such as PQL (Li et al., 2023b). However, it remains an open question whether greater inter-policy diversity necessarily translates into better performance.

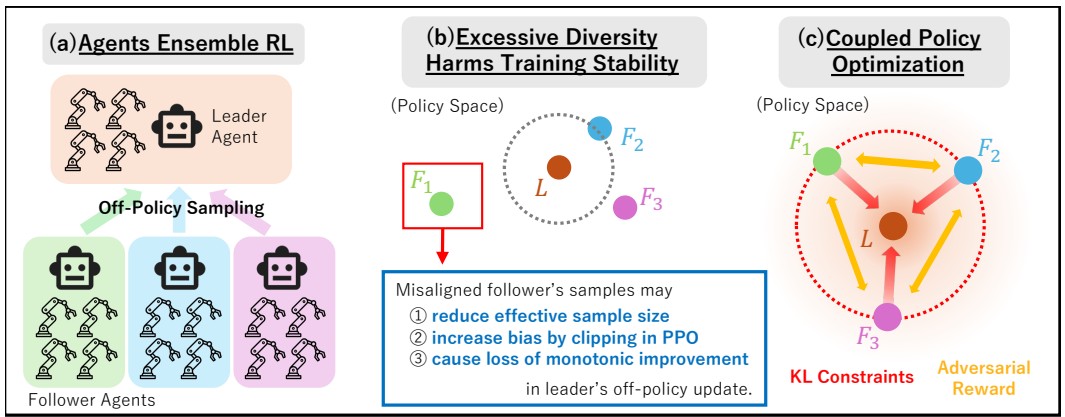

Figure 1: **Appropriately controlled policy diversity improves the learning efficiency of ensemble RL in large-scale environments.** (a) The leader-follower approach is an agent ensemble method that aggregates samples from multiple followers into a leader policy. (b) Misalignment between policies may causes a decline in sample efficiency and training stability. (c) Our method introduces KL divergence constraints to keep followers distributed around the leader, as well as adversarial reward to prevent policies overconcentration.

In this work, we theoretically and empirically investigate the impact of inter-policy diversity on ensemble policy gradient methods, showing that excessive diversity can harm both training stability and sample efficiency as shown in Fig. 1(b). To address this issue, we propose Coupled Policy Optimization (CPO), a novel method that introduces a KL divergence constraint during follower policy updates in the leader-follower framework, thereby promoting diverse yet well-structured exploration around the leader (Fig. 1(c)). In addition, to prevent overconcentration among policies, we incorporate an adversarial reward that discriminates agent identity from state–action pairs, ensuring balanced and effective diversity.

Extensive experiments on dexterous manipulation, gripper-based manipulation and locomotion tasks demonstrate that our method outperforms strong baselines such as SAPG, DexPBT, and PPO in both sample efficiency and final performance. In addition, we confirm that the KL constraint drives the IS ratios closer to one, which increases the effective sample size (ESS) and mitigates the clipping bias in PPO, thereby improving the effective sample efficiency and training stability. Furthermore, analysis of the ensemble policies reveals that SAPG suffers from severe policy misalignment, where some follower policies diverge significantly from the leader, hindering leaning ability. In contrast, CPO naturally induces a stable and well-structured policy formation, with followers distributed around the leader in a balanced manner.

To summarize, our main contributions are as follows:

- We provide a theoretical analysis showing that excessive inter-policy diversity in ensemble policy gradient methods degrades training stability and sample efficiency.
- We propose CPO, a leader-follower framework that introduces a KL divergence constraint and adversarial reward during follower updates to enable effective and stable exploration in policy space. The proposed method outperforms strong baselines including SAPG, DexPBT, and PPO across challenging robotic tasks.
- We empirically verify that the KL constraint keeps IS ratios close to one in leader's off-policy policy update, leading to improved sample efficiency.
- Through inter-policy KL divergence analysis, we show that CPO naturally induces a structured policy formation in which follower policies are consistently distributed around the leader policy, avoiding the policy misalignment observed in a prior method.

## 2 RELATED WORK

### 2.1 DISTRIBUTED REINFORCEMENT LEARNING

Deep reinforcement learning (RL) relies on trial-and-error, and increased data collection through massively parallel environments directly contributes to performance improvement. Early work focused on asynchronous distributed algorithms across multiple devices with hundreds to thousands of environments, favoring off-policy methods (Espeholt et al., 2018; Horgan et al., 2018; Espeholt et al., 2019). Recently, GPU-based simulators such as Isaac Gym (Makoviychuk et al., 2021) have enabled tens of thousands of environments to run synchronously on a single device, reviving interest in on-policy methods that often achieve higher final performance in robotic tasks (Rudin et al., 2022; Handa et al., 2023; Zhuang et al., 2023; Li et al., 2023b). However, naively scaling methods like PPO to such large numbers of environments yields diminishing returns, since a single policy provides limited exploration diversity, resulting in similar trajectories (Singla et al., 2024).

### 2.2 AGENT ENSEMBLE IN PARALLELED ENVIRONMENTS

To enhance exploration diversity in massively parallel environments, ensemble methods with multiple policies have been explored. DexPBT (Aleksei Petrenko, 2023), for example, trains policies with different hyperparameters in parallel but discards data from non-selected policies, reducing overall efficiency. SAPG (Singla et al., 2024) instead leverages all follower data through IS in a leader-follower framework, improving exploration diversity and training stability. Yet, the impact of inter-policy diversity has not been thoroughly examined, and excessively divergent followers may generate off-policy samples that destabilize the leader.

### 2.3 POLICY UPDATE WITH REGULARIZATION

Policy regularization is widely used in RL, typically constraining divergence either from the dataset policy in offline RL (Fujimoto & Gu, 2021; Garg et al., 2023; Sikchi et al.; Nair et al., 2020) or from the old policy in online RL (Schulman et al., 2015; 2017; Abdolmaleki et al., 2018), thereby improving stability and efficiency. Beyond being used purely as constraints, policy regularization has also been extended to explicitly promote diversity across multiple policies, using behavioral or distributional distance metrics (Parker-Holder et al., 2020; Yao et al., 2023; Wu et al., 2022). In this work, we regularize the divergence between follower and leader policies so that followers explore near the leader in policy space, collecting data informative to the leader while maintaining diversity. For this purpose, and motivated by our theoretical analysis of the leader policy's learning dynamics, we employ KL divergence, following prior approaches such as XQL (Garg et al., 2023) and AWAC (Nair et al., 2020).

## 3 PRELIMINARIES

In this paper, we theoretically show that excessive inter-policy diversity in ensemble policy gradient methods under massively parallel environments can harm training stability and sample efficiency, and we propose a method that controls the diversity between agents to promote efficient exploration. Since both our analysis and the proposed method build upon the leader-follower framework of SAPG (Singla et al., 2024), we first review the formulation of a fundamental on-policy algorithm, PPO (Schulman et al., 2017), and then summarize the key ideas and limitations of SAPG.

### 3.1 REINFORCEMENT LEARNING

In RL, tasks are typically formalized as a Markov Decision Process (MDP), defined by a tuple $(\mathcal{S}, \mathcal{A}, P, r, \gamma, d)$. Here, $\mathcal{S}$ is the state space, $\mathcal{A}$ is the action space, $P(\boldsymbol{s}_{t+1}|\boldsymbol{s}_t, \boldsymbol{a}_t)$ is the state transition probability density, $r(\boldsymbol{s}, \boldsymbol{a})$ is the reward function, $\gamma$ is the discount factor, and $d(\boldsymbol{s}_0)$ is the initial state distribution. A policy $\pi(\boldsymbol{a}|\boldsymbol{s}) : \mathcal{S} \times \mathcal{A} \mapsto \mathbb{R}$ is defined as a probability distribution over actions conditioned on the state. The objective of RL is to learn a policy that maximizes the expected return $\mathbb{E}[R_0|\pi]$ where $R_t = \Sigma_{k=t}^{T} \gamma^{k-t} r(\boldsymbol{s}_k, \boldsymbol{a}_k)$ and $T$ is a task horizon.

## 3.2 Proximal Policy Optimization (PPO)

PPO is a widely used on-policy algorithm that stabilizes updates by clipping the IS ratio with the behavior policy. All agents in this study are trained with PPO with modifications. The objective is:

$$L_{\text{PPO}}(\theta) = -\mathbb{E}_{\boldsymbol{s},\boldsymbol{a}\sim\pi_{\theta_{\text{old}}}} \left[ \min(r(\theta)A(\boldsymbol{s},\boldsymbol{a}), \text{clip}(r(\theta), 1-\epsilon, 1+\epsilon)A(\boldsymbol{s},\boldsymbol{a})) \right], \quad (1)$$

where $r(\theta) = \frac{\pi_\theta(a|s)}{\pi_{\theta_{\text{old}}}(a|s)}$ is the IS ratio and $\epsilon$ is the clipping parameter. The advantage function is $A(\boldsymbol{s},\boldsymbol{a}) = Q^{\pi_\theta}(\boldsymbol{s},\boldsymbol{a}) - V^{\pi_\theta}(\boldsymbol{s})$, with the action-value function $Q^{\pi_\theta}(\boldsymbol{s},\boldsymbol{a}) = \mathbb{E}_{\pi_\theta}[R \mid \boldsymbol{s},\boldsymbol{a}]$ and the value function $V^{\pi_\theta}(\boldsymbol{s}) = \mathbb{E}_{\pi_\theta}[R \mid \boldsymbol{s}]$. Thus, $A(\boldsymbol{s},\boldsymbol{a})$ measures how much better action $\boldsymbol{a}$ is compared to the average action under $\pi_\theta$.

## 3.3 Split and Aggregate Policy Gradients (SAPG)

SAPG is a state-of-the-art RL method designed to enhance exploration diversity and sample efficiency in massively parallel environments. It trains multiple policies concurrently, where each follower agent collects data that is aggregated into a leader policy. The leader leverages off-policy data from followers through IS, enabling diverse exploration with parallel environments.

Specifically, the $N$ parallel environments are divided into $M$ blocks, and one leader policy and $M-1$ follower policies are each assigned to the blocks. All agents share the same policy and value networks conditioned on identification vectors. The leader policy $\pi_{L_\theta}(\boldsymbol{a}|\boldsymbol{s})$ and follower policies $\pi_{F_{i,\theta}}(\boldsymbol{a}|\boldsymbol{s})$, where $i \in \{0, \ldots, M-2\}$, are updated by the objective functions in Eq. 2 and Eq. 3.

$$L_{\text{SAPG},L}(\theta,j) = -\mathbb{E}_{\boldsymbol{s},\boldsymbol{a}\sim\pi_{L_{\theta_{\text{old}}}}} \left[ \min\left( r_{L_{\text{on}}}(\theta)A^L(\boldsymbol{s},\boldsymbol{a}), \text{clip}(r_{L_{\text{on}}}(\theta), 1-\epsilon, 1+\epsilon)A^L(\boldsymbol{s},\boldsymbol{a}) \right) \right]$$
$$- \mathbb{E}_{\boldsymbol{s},\boldsymbol{a}\sim\pi_{F_{j,\theta_{\text{old}}}}} \left[ \min\left( r_{L_{\text{off}}}(\theta)A^L(\boldsymbol{s},\boldsymbol{a}), \text{clip}(r_{L_{\text{off}}}(\theta), 1-\epsilon, 1+\epsilon)A^L(\boldsymbol{s},\boldsymbol{a}) \right) \right], \quad (2)$$

$$L_{\text{SAPG},F_i}(\theta) = -\mathbb{E}_{\boldsymbol{s},\boldsymbol{a}\sim\pi_{F_{i,\theta_{\text{old}}}}} \left[ \min\left( r_{F_i}(\theta)A^{F_i}(\boldsymbol{s},\boldsymbol{a}), \text{clip}(r_{F_i}(\theta), 1-\epsilon, 1+\epsilon)A^{F_i}(\boldsymbol{s},\boldsymbol{a}) \right) \right], \quad (3)$$

where $j \in \{0, \ldots, M-2\}$ denotes the index of a follower agent randomly sampled at each training epoch, and the density ratios between behavior policy and the updating policy are defined as:

$$r_{L_{\text{on}}}(\theta) = \frac{\pi_{L_\theta}(\boldsymbol{a}|\boldsymbol{s})}{\pi_{L_{\theta_{\text{old}}}}(\boldsymbol{a}|\boldsymbol{s})}, \quad r_{L_{\text{off}}}(\theta) = \frac{\pi_{L_\theta}(\boldsymbol{a}|\boldsymbol{s})}{\pi_{F_{j,\theta_{\text{old}}}}(\boldsymbol{a}|\boldsymbol{s})}, \quad r_{F_i}(\theta) = \frac{\pi_{F_{i,\theta}}(\boldsymbol{a}|\boldsymbol{s})}{\pi_{F_{i,\theta_{\text{old}}}}(\boldsymbol{a}|\boldsymbol{s})}. \quad (4)$$

Here, $A^{F_i}(\boldsymbol{s},\boldsymbol{a})$ and $A^L(\boldsymbol{s},\boldsymbol{a})$ denote the advantage functions for the $i$-th follower and the leader policy, respectively. Furthermore, SAPG introduces an entropy regularization term applied to all policies to encourage diversity in exploration across agents.

However, SAPG lacks an explicit mechanism to control the distance between leader and follower policies while applying entropy regularization, which may cause followers to drift significantly from the leader. As a preliminary study, we conducted an ablation on the entropy regularization term in SAPG and found that, although it promotes exploration and can improve sample efficiency, it also increases the follower–leader KL divergence and often leads to severe misalignment. See Appendix A.7 for details. In this paper, we analyze how such excessive divergence affects learning.

## 4 Effect of Policy Diversity on Ensemble Policy Gradient

Policy diversity affects ensemble policy gradient methods in two major aspects: data coverage and training stability. While diverse exploration increases coverage and mitigates local optima, excessive diversity reduces sample density and weakens the variance-reduction effect of parallel environments, reflecting a fundamental exploration–exploitation trade-off in reinforcement learning. More critically, excessive divergence between the leader and follower policies can directly harm training stability and sample efficiency. We formalize this intuition through the following propositions.

> **Proposition 1.** *The expected absolute deviation of the IS ratio from 1 is inversely related to the effective sample size (ESS); as the deviation increases, the ESS decreases.*

When the leader and follower policies diverge, the expected absolute deviation of the IS ratio for leader update with follower samples, $\mathbb{E}_{\boldsymbol{s},\boldsymbol{a}\sim\pi_{F_{\text{old}}}}\left[\left|1 - \frac{\pi_L(\boldsymbol{a}|\boldsymbol{s})}{\pi_{F_{\text{old}}}(\boldsymbol{a}|\boldsymbol{s})}\right|\right]$, increases. This deviation leads to higher variance in the IS ratio, thereby diminishing the ESS, which is a standard metric of sample efficiency in IS with approximation (Martino et al., 2017), where $w_i$ is the IS ratio, defined as follows:

$$ESS = \frac{1}{\sum_{i=1}^{N} \tilde{w}_i^2}, \quad \tilde{w}_i = \frac{w_i}{\sum_{j=1}^{N} w_j}. \tag{5}$$

Intuitively, samples from misaligned follower policies contribute little to the leader's learning, thereby reducing the overall sample efficiency of the leader update. The detailed derivation is provided in Appendix A.1.1.

> **Proposition 2.** *The $L^2$ norm of the bias of the gradient estimate induced by the PPO clipping operator is upper bounded by the square root of an expectation involving the IS ratio deviation.*

PPO ensures learning stability by clipping the IS ratio, however, this introduces bias into the gradient estimate. As the IS deviation increases, the effect of clipping becomes more pronounced, resulting in larger bias and destabilizing the leader's learning. This can be shown by upper bounding the $L^2$ norm of the bias as a function of the IS deviation. The detailed derivation is provided in Appendix A.1.2.

These propositions show that while policy diversity improves exploration, excessive divergence between the leader and follower policies causes the IS ratio to deviate from 1, which may undermine the sample efficiency and stability of the leader update by reducing ESS and increasing the gradient estimation bias, as shown in Fig. 1(b). We then examine how this deviation can be suppressed.

> **Proposition 3.** *For the leader update with follower samples, The expected absolute deviation of IS ratio from 1 is upper bounded by the KL divergence between the follower and leader policies.*

*Proof.* From Pinsker's inequality, we have $\|P - Q\| \leq \sqrt{2D_{\text{KL}}(P\|Q)}$ for any two distributions $P$ and $Q$. Applying this to the leader and follower policies, then:

$$\int_{\boldsymbol{a}} |\pi_{F_{\text{old}}}(\boldsymbol{a}|\boldsymbol{s}) - \pi_L(\boldsymbol{a}|\boldsymbol{s})|\, d\boldsymbol{a} \leq \sqrt{2D_{\text{KL}}(\pi_{F_{\text{old}}}(\cdot|\boldsymbol{s})\|\pi_L(\cdot|\boldsymbol{s}))}. \tag{6}$$

Here, using the identity $\left|1 - \frac{\pi_L(\boldsymbol{a}|\boldsymbol{s})}{\pi_{F_{\text{old}}}(\boldsymbol{a}|\boldsymbol{s})}\right| = \left|\frac{\pi_{F_{\text{old}}}(\boldsymbol{a}|\boldsymbol{s}) - \pi_L(\boldsymbol{a}|\boldsymbol{s})}{\pi_{F_{\text{old}}}(\boldsymbol{a}|\boldsymbol{s})}\right|$, we take the expectation with respect to $\boldsymbol{a} \sim \pi_{F_{\text{old}}}$:

$$\mathbb{E}_{\boldsymbol{a}\sim\pi_{F_{\text{old}}}}\left[\left|1 - \frac{\pi_L(\boldsymbol{a}|\boldsymbol{s})}{\pi_{F_{\text{old}}}(\boldsymbol{a}|\boldsymbol{s})}\right|\right] = \int_{\boldsymbol{a}} \pi_{F_{\text{old}}}(\boldsymbol{a}|\boldsymbol{s}) \left|\frac{\pi_{F_{\text{old}}}(a|s) - \pi_L(\boldsymbol{a}|\boldsymbol{s})}{\pi_{F_{\text{old}}}(\boldsymbol{a}|\boldsymbol{s})}\right| d\boldsymbol{a} \tag{7}$$

$$\leq \sqrt{2D_{\text{KL}}(\pi_{F_{\text{old}}}(\cdot|\boldsymbol{s})\|\pi_L(\cdot|\boldsymbol{s}))}. \tag{8}$$

Furthermore, assuming reachability, we take the expectation over the states encountered by the follower policy. This yields $\mathbb{E}_{\boldsymbol{s},\boldsymbol{a}\sim\pi_{F_{\text{old}}}}\left[\left|1 - \frac{\pi_L(\boldsymbol{a}|\boldsymbol{s})}{\pi_{F_{\text{old}}}(\boldsymbol{a}|\boldsymbol{s})}\right|\right] \leq \sqrt{2D_{\text{KL}}(\pi_{F_{\text{old}}}(\cdot|\boldsymbol{s})\|\pi_L(\cdot|\boldsymbol{s}))}$, showing that as the KL divergence increases, the IS ratio deviates further from 1. $\square$

Consequently, introducing a constraint on the KL divergence between the leader and follower policies alleviates the IS ratio deviation. Schulman et al. (2015) and Xie et al. (2025) also argue that as long as the KL divergence or IS ratio deviation between the target and behavior policies remains small, the update error due to distribution shift is reduced, and performance improvement is guaranteed. These motivate the need for KL-based coupling between leader and followers, to regulate policy diversity in ensemble policy gradient methods.

## 5 COUPLED POLICY OPTIMIZATION

Building upon the theoretical observation in section 4, we propose CPO, a method that regulates the inter-agent distance during training. Our approach extends SAPG (Singla et al., 2024) by constraining the KL divergence between the leader and each follower policy during follower updates, enabling diverse yet meaningful exploration for the leader. Furthermore, we introduce an auxiliary adversarial reward that encourages diversity across follower policies, to prevent overconcentration of agents.

### 5.1 FOLLOWER'S POLICY UPDATE UNDER KL CONSTRAINT

We formulate the update of each follower policy as a constrained optimization problem with a KL divergence constraint to the leader policy:

$$\pi_{F_i}^*(\boldsymbol{a}|\boldsymbol{s}) = \arg\max_{\pi_{F_i}} A_{F_i}(\boldsymbol{s}, \boldsymbol{a}) \quad \text{s.t. } D_{\mathrm{KL}}(\pi_{F_i}(\cdot|\boldsymbol{s}) \,\|\, \pi_L(\cdot|\boldsymbol{s})) \leq \varepsilon_{\mathrm{KL}}. \tag{9}$$

Following the approach of AWAC (Nair et al., 2020), this problem admits a closed-form non-parametric solution, which we then approximate with a neural network policy $\pi_{F_{i,\theta}}(a|s)$. The resulting parametric objective of follower update can be written as follows:

$$L_{\mathrm{CPO},F_i}(\theta) = -\mathbb{E}_{\boldsymbol{a},\boldsymbol{s} \sim \pi_{L_{\theta_{\mathrm{old}}}}} \left[ \log \pi_{F_{i,\theta}}(\boldsymbol{a}|\boldsymbol{s}) \exp\left(\frac{1}{\lambda_f} A^{F_i}(\boldsymbol{s}, \boldsymbol{a})\right) \right] + L_{\mathrm{SAPG},F_i}(\theta), \tag{10}$$

where, $\lambda_f$ is a temperature parameter to control the strength of KL constraint. The detailed derivation of Eq. 10 is provided in Appendix A.2. Thus, the policy objective of our proposed method, $L_{\mathrm{CPO}}(\theta, j)$, can be expressed as an extension of the SAPG policy objective $L_{\mathrm{SAPG}}(\theta, j)$ as:

$$L_{\mathrm{CPO}}(\theta) = L_{\mathrm{SAPG}}(\theta, j) + \beta \sum_{i \in \{0,\dots,M-2\}} L_{\mathrm{CPO},F_{i,f}}(\theta, \lambda_f), \tag{11}$$

where $\beta$ is a coefficient introduced to roughly match the scale between the PPO objective and the KL-regularized loss term, which involves an exponential. The pseudocode of our method and the discussion on computational complexity are provided in Appendix A.3.

### 5.2 ADVERSARIAL REWARD FOR FOLLOWERS DISTRIBUTION

In our method, follower policies are trained to explore within a KL-bounded neighborhood around the leader to preserve the stability of the leader's off-policy PPO update. While it prevents harmful misalignment issue, it also indirectly pulls followers closer to one another, which can reduce the diversity of their state–action coverage within the neighborhood.

To mitigate the issue, we introduce an intrinsic reward to encourage sufficient separation among the policies. Although there are various ways to promote diverse exploration, e.g. RDN (Burda et al., 2018) and ICM (Pathak et al., 2017), they are not intended for scenarios where multiple policies perform rollouts in parallel and do not explicitly account for the distance between policies.

Therefore, we draw inspiration from DIAYN (Eysenbach et al., 2018), which explicitly encourage separation between policies. We train a discriminator $D_\xi(y|\boldsymbol{s}_t, \boldsymbol{a}_t)$, parameterized by a neural network with parameters $\xi$, to predict the index $y \in \{0, \dots, M-1\}$ of the policy given a state-action pair and the classification loss is then used as an intrinsic reward. This encourages each follower to explore distinct regions in the state-action space, such that the discriminator can identify their identity . Given a data buffer $\mathcal{D}$, the discriminator loss and the intrinsic reward are given by:

$$L_D(\xi) = -\mathbb{E}_{(\boldsymbol{s}_t,\boldsymbol{a}_t,y) \sim \mathcal{D}}[\log D_\xi(y|\boldsymbol{s}_t, \boldsymbol{a}_t)], \quad r_t^{\mathrm{adv}}(\boldsymbol{s}_t, \boldsymbol{a}_t, y) = \lambda_{\mathrm{adv}} \log D_\xi(y|\boldsymbol{s}_t, \boldsymbol{a}_t). \tag{12}$$

Notably, this intrinsic reward is not provided to the leader agent. When the leader is updated from the off-policy samples collected by followers, only the true environment rewards are considered.

## 6 EXPERIMENTS

We evaluated our method on six dexterous manipulation tasks (Andrychowicz et al., 2020; Aleksei Petrenko, 2023), those have high dimensional action space, two gripper-based manipulation tasks, and two locomotion tasks to compare its performance against state-of-the-art methods under massively parallel settings. All tasks provide dense rewards and the experiments were conducted on Isaac Gym (Makoviychuk et al., 2021) with $N = 24,576$ parallel environments, following the experimental setup of the prior work (Singla et al., 2024). Detailed descriptions of the tasks are provided in Appendix A.6.

For baselines, we selected PPO, DexPBT (Aleksei Petrenko, 2023), and SAPG (Singla et al., 2024). All of these methods are built upon PPO.

- **PPO** (Schulman et al., 2017): is a representative policy gradient algorithm widely used across various tasks. We simply increased the number of samples collected per epoch to be equal to the product of the horizon length and the number of environments $N$.

- **DexPBT** (Aleksei Petrenko, 2023): is a population-based parallel learning framework that divides the $N$ environments into $M$ subsets, where $M$ agents each with different hyperparameters train in parallel. Periodically, the lowest-performing agents are removed and replaced by new agents generated through genetic algorithms, which assign updated hyperparameters for the next training phase.

- **SAPG** (Singla et al., 2024): adopts agent ensemble learning based on a leader-follower network, and it represents the state-of-the-art in massively parallel environments to the best of our knowledge. The leader agent is updated with not only its own on-policy samples, but also off-policy samples from follower agents through IS.

For DexPBT, SAPG, and our proposed method, we set the number of parallel blocks to $M = 6$. In SAPG and our method, the shared networks are conditioned on a one-dimensional vector $\phi \in \mathbb{R}^1$. Hyperparameters common to both SAPG and our method, such as the entropy coefficient, were set to the same values as used in SAPG. The hyperparameters and computing environments used in all experiments are provided in Appendix A.6. All experiments were conducted using five random seeds.

## 7    RESULTS AND ANALYSIS

We analyze the learning performance of our proposed method compared to baselines, conduct an ablation study on the strength of the KL constraint, and examine the evolution of inter-policy KL divergence during training.

### 7.1    TRAINING PERFORMANCE

To compare the training performance of each method, we present the learning curves on each task in Fig.2, along with the final performance on dexterous manipulation tasks after $2 \times 10^{10}$ environment steps training summarized in Table1. Each result shows the mean and standard deviation over five random seeds.

Our proposed method consistently achieves high sample efficiency and strong final performance across all tasks especially on the manipulation tasks. In particular, while PBT fails to learn on the AllegroKuka Regrasping and Franka tasks, and SAPG struggles in the Two-Arms Reorientation task, our method demonstrates robust learning capability. Moreover, in many tasks, it reaches SAPG's final performance with approximately half the number of environment steps, indicating the acquisition of efficient exploration ability. No significant improvement over SAPG is observed in the AllegroKuka Regrasping and Throw tasks, which we discuss in the next section.

In locomotion tasks, due to their relative simplicity, the performance differences across algorithms are smaller. Nevertheless, PBT exhibits faster convergence, indicating that in simpler tasks, broad parallel exploration can be advantageous. On the other hand, our method converges slightly faster

Table 1: **Performance on dexterous manipulation tasks after $2 \times 10^{10}$ environment steps of training.** Bold indicates the method with the highest average performance for each task, as well as those not significantly different from it, as determined by a t-test (p > 0.05).

| Task | PPO | PBT | SAPG | CPO (ours) |
|---|---|---|---|---|
| ShadowHand | $10661 \pm 1050$ | $10294 \pm 1728$ | $12882 \pm 343$ | $\mathbf{13762 \pm 414}$ |
| AllegroHand | $10439 \pm 1282$ | $13239 \pm 239$ | $11989 \pm 817$ | $\mathbf{14421 \pm 885}$ |
| Regrasping | $0.76 \pm 0.99$ | $\mathbf{35.26 \pm 2.82}$ | $\mathbf{37.20 \pm 0.65}$ | $\mathbf{37.44 \pm 1.21}$ |
| Reorientation | $1.04 \pm 0.98$ | $2.92 \pm 4.27$ | $38.79 \pm 1.66$ | $\mathbf{43.75 \pm 0.65}$ |
| Throw | $15.69 \pm 3.34$ | $19.08 \pm 1.02$ | $\mathbf{22.51 \pm 1.15}$ | $\mathbf{21.69 \pm 2.44}$ |
| Two-Arms Reorientation | $1.41 \pm 0.80$ | $\mathbf{26.43 \pm 11.12}$ | $5.11 \pm 3.41$ | $\mathbf{35.30 \pm 2.77}$ |

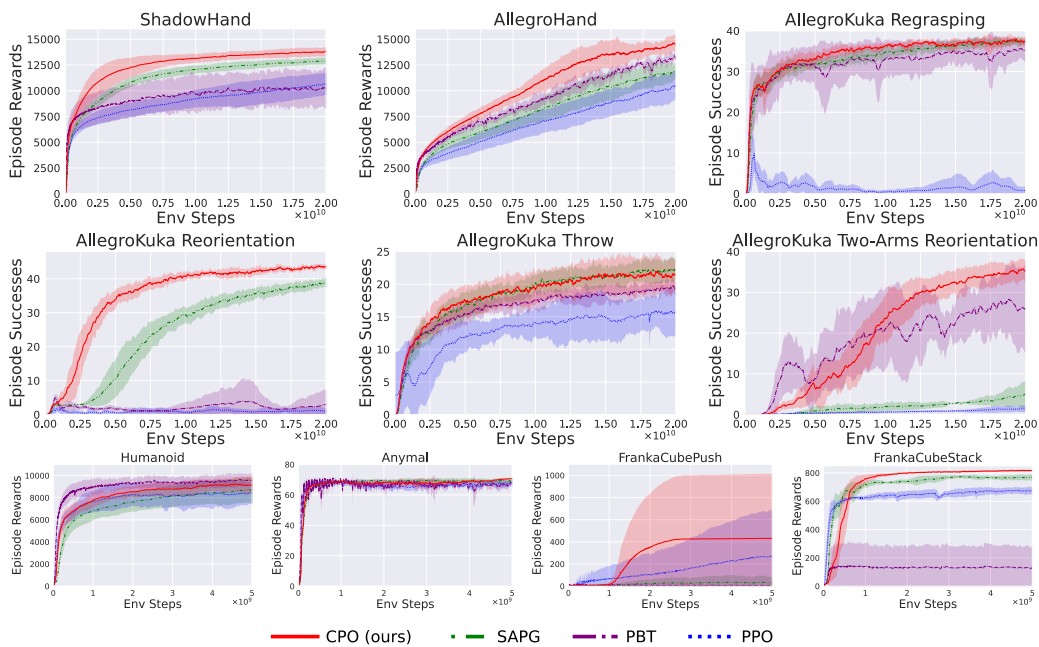

Figure 2: **Comparison of algorithm performance across ten robotic tasks.** Learning curves across six dexterous manipulation, two gripper-based manipulation and two locomotion tasks comparing CPO to SAPG, PBT, and PPO. CPO consistently achieves higher sample efficiency and final performance, particularly in ShadowHand, AllegroHand, AllegroKukaReorientation, Two-Arms Reorientation, FrankaCubePush and Stack.

than SAPG, suggesting that in leader-follower policy gradient frameworks, the stabilization and sample efficiency gains brought by KL constraints outweigh the benefits of broader data coverage through exploration diversity.

We also conducted an ablation study to isolate the contributions of the adversarial reward and the KL constraint, as shown in Appendix A.4.

## 7.2 ABLATION STUDY ON KL CONSTRAINT

To assess the sensitivity to the KL constraint hyperparameter and to empirically verify the propositions in section 4, we conducted an ablation study varying the KL coefficient ($\lambda_f$) in the Shadow Hand and AllegroKuka Reorientation tasks. To isolate the effect of the KL constraint, we conducted experiments without the adversarial reward. Also, $\beta$ in Eq. 11 was fixed at $0.001$.

Fig. 3 presents training curves of our method with different $\lambda_f$ values compared to SAPG, demonstrating that CPO was robust to a wide range of values, consistently outperforming SAPG. A practical tuning heuristic is starting with a weak constraint ($\lambda_f = 0.5$) and gradually strengthen it.

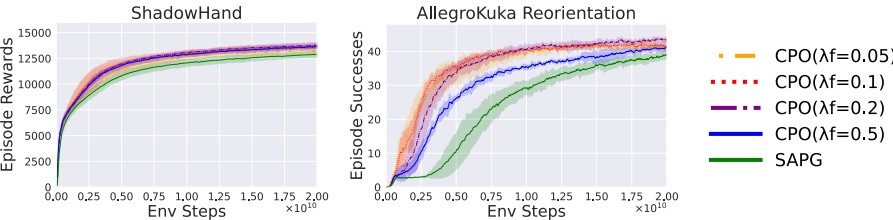

Figure 3: **Training Curves from the ablation study with different $\lambda_f$.**

Table 2: **Mean IS Ratio Deviation and Overall ESS Rate at $5 \times 10^9$ environment steps.** The reported values are computed by averaging over a window of eleven iterations.

| Task | Method | Mean IS Ratio Deviation ($\downarrow$) | ESS Rate ($\uparrow$) |
|---|---|---|---|
| ShadowHand | SAPG | 0.889 | 0.0223 |
| | CPO(0.5) | 0.403 | 0.763 |
| | CPO(0.2) | 0.297 | 0.871 |
| | CPO(0.1) | 0.222 | 0.923 |
| | CPO(0.05) | 0.187 | 0.941 |
| AllegroKuka Reorientation | SAPG | 0.608 | 0.110 |
| | CPO(0.5) | 0.420 | 0.721 |
| | CPO(0.2) | 0.276 | 0.888 |
| | CPO(0.1) | 0.214 | 0.929 |
| | CPO(0.05) | 0.199 | 0.938 |

Table 2 shows the mean IS ratio deviation from 1 and the ESS (normalized to a maximum of 1) at $5 \times 10^9$ environment steps. The deviation is computed from all follower samples, and the ESS from all leader and follower samples. Consistent with Proposition 1 in section 4, we observed that stronger KL constraints (smaller $\lambda_f$) lead to smaller deviations and higher ESS, improving sample efficiency.

### 7.3 KL Divergence Analysis

In this section, we analyze the KL divergence between policies during training to compare agent relationships in SAPG and our method (Fig. 4; higher-resolution results are provided in Appendix A.5). In ShadowHand and AllegroKuka Reorientation, where our method clearly outperformed SAPG, several SAPG followers misaligned significantly from the leader, producing harmful samples that hinder the leader's learning, as described in section 4. In contrast, our method maintained stable inter-agent distances, yielding more informative samples. In AllegroKuka Regrasping, where both methods achieved similar performance, SAPG followers did not show noticeable divergence, likely due to incidental alignment between SAPG's shared backbone and the task characteristics.

Interestingly, in our method the leader consistently remained the closest agent to every follower (white circles in Fig. 4), suggesting that KL regularization, together with adversarial reward and entropy terms, naturally distributes followers around the leader without overconcentration. Furthermore, unlike SAPG's ablation where all agents sampled from each other, leading to excessive similarity and reduced diversity (Singla et al., 2024), our approach preserves the leader-follower asymmetry: each follower learns only from its own on-policy data and the leader's off-policy data. This design helps maintain diversity while keeping inter-policy distances under control.

## 8 Conclusion and Limitation

In this work, we theoretically showed that excessive inter-policy diversity in ensemble policy gradient methods under massively parallel environments can harm sample efficiency and stability by reducing effective sample size, increasing clipping bias, and weakening monotonic improvement guarantees. To address this issue, we proposed Coupled Policy Optimization, which introduces KL constraints between leader and follower policies and adversarial rewards to prevent overconcentration. Experiments on dexterous manipulation, gripper-based manipulation and locomotion tasks demonstrated that CPO outperforms strong baselines such as SAPG, PBT, and PPO in both sample efficiency and final performance. Ablation studies confirmed that KL constraint reduces IS-ratio deviation and improves effective sample size, while KL-divergence visualizations revealed that followers naturally distribute around the leader without misalignment, highlighting the stability and structural effectiveness of our method.

These findings suggest that in ensemble policy gradient methods under massively parallel environments, it is not sufficient to merely promote policy diversity; rather, appropriate control of diversity is crucial for achieving both stable and sample-efficient learning.

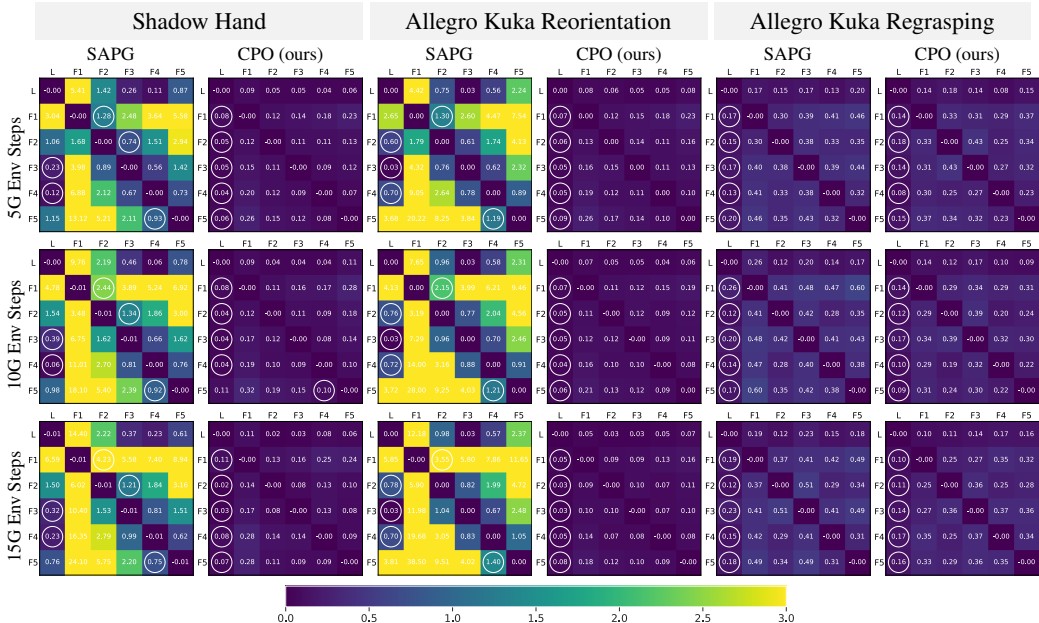

Figure 4: **Comparison of the transition of KL divergence between agents with different algorithms.** Each heatmap shows the KL divergence between the leader and follower policies during training. Row $i$, column $j$ indicates the forward KL from agent $i$ to agent $j$. The white circle marks the agent closest from each follower, excluding itself. SAPG often shows misaligned followers, while our method keeps them well-distributed around the leader.

A limitation of our method is still rely on a fixed number of policies and environments per policy. However, the effective exploration range can vary with the task and training stage. Developing algorithms that automatically adjust these parameters would be an interesting future direction, unlocking the potential of massively parallel environments, especially for tasks with high-dimensional action spaces and demanding exploration requirements.

## ACKNOWLEDGEMENT

This work was partially supported by JST Moonshot R&D Grant Number JPMJPS2011, CREST Grant Number JPMJCR2015, Basic Research Grant (Super AI) of Institute for AI and Beyond of the University of Tokyo, and the IIW program of the University of Tokyo. N.S. was supported by Hayashi Rheology Memorial Foundation,Utsunomiya, Japan. T.O. was supported by JSPS KAKENHI Grant Number JP25K03176.

## REPRODUCIBILITY STATEMENT

To facilitate reproducibility, we provide the source code of our proposed method, CPO, at the following repository `https://github.com/Naoki04/paper-cpo-code`. The accompanying README highlights the files that contain the key functions used in our implementation. Details of the experimental environments, as well as the hyperparameters used in all experiments are listed in Appendix A.6.

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

# A APPENDIX

## A.1 PROOFS OF PROPOSITIONS

In this section, we provide the proofs of Proposition 1 and Proposition 2 stated in section 4.

### A.1.1 PROOF OF PROPOSITION 1

*Proof.* Let $N_{L,\text{on}}$ denote the number of leader (on-policy) samples and $N_{L,\text{off}}$ the number of follower (off-policy) samples. Assuming reachability, i.e., the support of the target policy is contained in that of the behavior policy, the IS ratio has unit mean: $\mathbb{E}[r] = 1$. Then, ESS for the leader update can be

expressed as a function of $\text{Var}_{s,a\sim\pi_{F_{\text{old}}}}[r_{L,\text{off}}(\theta)]$ as follows:

$$ESS = \frac{\left(\sum_{i=1}^{N_{L,\text{on}}+N_{L,\text{off}}} w_i\right)^2}{\sum_{i=1}^{N_{L,\text{on}}+N_{L,\text{off}}} w_i^2}, \tag{13}$$

$$= \frac{\left(N_{L,\text{on}}\mathbb{E}_{s,a\sim\pi_{L_{\text{old}}}}[r_{L,\text{on}}(\theta)] + N_{L,\text{off}}\mathbb{E}_{s,a\sim\pi_{F_{\text{old}}}}[r_{L,\text{off}}(\theta)]\right)^2}{(\text{Var}_{s,a\sim\pi_{L_{\text{old}}}}[r_{L,\text{on}}(\theta)] + 1) + (\text{Var}_{s,a\sim\pi_{F_{\text{old}}}}[r_{L,\text{off}}(\theta)] + 1)}, \tag{14}$$

$$= \frac{(N_{L,\text{on}} + N_{L,\text{off}})^2}{\text{Var}_{s,a\sim\pi_{L_{\text{old}}}}[r_{L,\text{on}}(\theta)] + \text{Var}_{s,a\sim\pi_{F_{\text{old}}}}[r_{L,\text{off}}(\theta)] + 2}. \tag{15}$$

Here, the variance of IS ratio for off-policy samples is lower bounded by the expected absolute deviation of it from 1 as:

$$\mathbb{E}_{s,a\sim\pi_{F_{\text{old}}}}[|1 - r_{L,\text{off}}|] \le \sqrt{\mathbb{E}_{s,a\sim\pi_{F_{\text{old}}}}[(1 - r_{L,\text{off}}(\theta))^2]}, \tag{16}$$

$$= \sqrt{\mathbb{E}_{s,a\sim\pi_{F_{\text{old}}}}[r_{L,\text{off}}(\theta)^2] - 2\mathbb{E}_{s,a\sim\pi_{F_{\text{old}}}}[r_{L,\text{off}}(\theta)] + 1}, \tag{17}$$

$$= \sqrt{\text{Var}_{s,a\sim\pi_{F_{\text{old}}}}[r_{L,\text{off}}(\theta)] + (\mathbb{E}_{s,a\sim\pi_{F_{\text{old}}}}[r_{L,\text{off}}(\theta)] - 1)^2}, \tag{18}$$

$$= \sqrt{\text{Var}_{s,a\sim\pi_{F_{\text{old}}}}[r_{L,\text{off}}(\theta)]}. \tag{19}$$

$\square$

### A.1.2 PROOF OF PROPOSITION 2

*Proof.* Formally, the policy gradient $g_{\text{ac}}$ and gradient estimated by PPO $g_{\text{ppo}}$ for behavior policy $\pi_{\theta,\text{old}}(a|s)$ and the target policy $\pi_\theta(a|s)$ can be expressed as:

$$g_{\text{ac}} = \mathbb{E}_{\pi_{\text{old}}}[r(\theta) \cdot \nabla_\theta \log \pi_\theta(a|s)A(s,a)], \tag{20}$$

$$g_{\text{ppo}} = \nabla_\theta \mathcal{L}_{\text{ppo}}, \tag{21}$$

where $r(\theta) = \frac{\pi_\theta(a|s)}{\pi_{\theta,\text{old}}(a|s)}$. Then, the gradient estimation bias introduced by PPO clipping operator can be written as follows:

$$\text{Bias} = g_{\text{ac}} - g_{\text{ppo}},$$
$$= \mathbb{E}_{s,a\sim\pi_{\theta,\text{old}}}[\nabla_\theta \log \pi_\theta(a|s)\, r(\theta)A(s,a)\,\mathbb{1}_{\text{clipped}}]. \tag{22}$$

Here, $\mathbb{1}_{\text{clipped}}$ denotes the indicator function that takes the value 1 if the PPO objective is in the clipped regime, and 0 otherwise. Therefore, the squared $L^2$ norm of this bias can be bounded using Jensen's inequality:

$$\|\text{Bias}\|_2^2 = \left\|\mathbb{E}_{s,a\sim\pi_{\theta,\text{old}}}[\nabla_\theta \log \pi_\theta(a|s)\, r(\theta)A(s,a)\,\mathbb{1}_{\text{clipped}}]\right\|^2,$$
$$\le \mathbb{E}_{s,a\sim\pi_{\theta,\text{old}}}\left[\|\nabla_\theta \log \pi_\theta(a|s)\|^2 \cdot r(\theta)^2 \cdot A(s,a)^2 \cdot \mathbb{1}_{\text{clipped}}\right],$$
$$\le \mathbb{E}_{s,a\sim\pi_{\theta,\text{old}}}\left[\|\nabla_\theta \log \pi_\theta(a|s)\|^2 \cdot r(\theta)^2 \cdot A(s,a)^2 \cdot \mathbb{1}_{|1-r(\theta)|>\epsilon}\right],$$
$$\le \mathbb{E}_{s,a\sim\pi_{\theta,\text{old}}}\left[\|\nabla_\theta \log \pi_\theta(a|s)\|^2 \cdot (|1-r(\theta)|+1)^2 \cdot A(s,a)^2 \cdot \mathbb{1}_{|1-r(\theta)|>\epsilon}\right], \tag{23}$$

where $\mathbb{1}_{|1-r(\theta)|>\epsilon}$ denotes the indicator function that takes the value 1 if $|1 - r(\theta)| > \epsilon$, and 0 otherwise. Thus, the upper bound of the bias norm depends directly on $|1 - r(\theta)|$, which increases as the IS ratio deviates from 1, leading training instability.

The same reasoning applies to the leader's off-policy updates using follower samples. In our framework, $\pi_{\theta,\text{old}}(a|s) = \pi_{F_{\text{old}}}(a|s)$ and $\pi_{\theta,\text{old}}(a|s) = \pi_L(a|s)$. Therefore, the upper bound on the bias norm of the leader's gradient estimate from follower's samples depends on $|1 - r_{L,\text{off}}(\theta)|$.

$\square$

## A.2 DERIVATION OF FOLLOWER POLICY UPDATE UNDER KL CONSTRAINT

This section presents the derivation of the follower policy objective in Eq. 10 under the proposed KL constraint. The constrained optimization problem shown in Eq. 9 has a closed-form solution, which can be obtained using the method of Lagrange multipliers, as follows:

$$\pi_{F_i}^*(\boldsymbol{a}|\boldsymbol{s}) = \frac{1}{Z} \pi_L(\boldsymbol{a}|\boldsymbol{s}) \exp\left(\frac{1}{\lambda} A^{F_i}(\boldsymbol{s}, \boldsymbol{a})\right), \tag{24}$$

where $Z = \int \pi_L(\boldsymbol{a}|\boldsymbol{s}) \exp\left(\frac{A^{F_i}(\boldsymbol{s},\boldsymbol{a})}{\lambda}\right) \mathrm{d}\boldsymbol{a}$ and $\lambda$ is the Lagrange multiplier associated with the KL constraint, which also serves as a temperature parameter controlling the strength of attraction between the leader and follower policies.

Since the closed-form solution is expressed in a non-parametric form, we approximate it using a neural network policy $\pi_{F_{i,\theta}}(\boldsymbol{a}|\boldsymbol{s})$. To this end, we formulate the problem of approximating the non-parametric solution with a parametric model as the minimization of both the forward and reverse KL divergences between them. The minimization of the forward KL divergence can be expressed as follows:

$$\arg\min_\theta D_{\mathrm{KL}}(\pi_{F_i}^*(\cdot|\boldsymbol{s})||\pi_{F_{i,\theta}}(\cdot|\boldsymbol{s}))$$

$$= \arg\min_\theta \int \pi_{F_i}^*(\boldsymbol{a}|\boldsymbol{s}) \log \frac{\pi_{F_i}^*(\boldsymbol{a}|\boldsymbol{s})}{\pi_{F_{i,\theta}}(\boldsymbol{a}|\boldsymbol{s})} \mathrm{d}\boldsymbol{a},$$

$$= \arg\min_\theta \int -\pi_L(\boldsymbol{a}|\boldsymbol{s}) \exp\left(\frac{1}{\lambda_f} A^{F_i}(\boldsymbol{s}, \boldsymbol{a})\right) \log \pi_{F_{i,\theta}}(\boldsymbol{a}|\boldsymbol{s}) \mathrm{d}\boldsymbol{a},$$

$$= \arg\min_\theta -\mathbb{E}_{\boldsymbol{s},\boldsymbol{a}\sim\pi_L} \left[\log \pi_{F_{i,\theta}}(\boldsymbol{a}|\boldsymbol{s}) \exp\left(\frac{1}{\lambda_f} A^{F_i}(\boldsymbol{s}, \boldsymbol{a})\right)\right]. \tag{25}$$

Here, the objective function is computed as the expectation with respect to the leader's off-policy samples. In contrast, the minimization of the reverse KL divergence can be written as follows:

$$\arg\min_\theta D_{\mathrm{KL}}(\pi_{F_{i,\theta}}(\cdot|\boldsymbol{s})||\pi_{F_i}^*(\cdot|\boldsymbol{s}))$$

$$= \arg\min_\theta \int \pi_{F_{i,\theta}}(\boldsymbol{a}|\boldsymbol{s}) \log \frac{\pi_{F_{i,\theta}}(\boldsymbol{a}|\boldsymbol{s})}{\pi_{F_i}^*(\boldsymbol{a}|\boldsymbol{s})} \mathrm{d}\boldsymbol{a},$$

$$= \arg\min_\theta \int \pi_{F_{i,\theta}}(\boldsymbol{a}|\boldsymbol{s}) \left(\log \pi_{F_{i,\theta}}(\boldsymbol{a}|\boldsymbol{s}) - \log \pi_L(\boldsymbol{a}|\boldsymbol{s}) - \frac{1}{\lambda_r} A^{F_i}(\boldsymbol{s}, \boldsymbol{a})\right) \mathrm{d}\boldsymbol{a},$$

$$= \arg\min_\theta -\mathbb{E}_{\boldsymbol{s},\boldsymbol{a}\sim\pi_{F_{i,\theta_{\mathrm{old}}}}} \left[\frac{\pi_{F_{i,\theta}}(\boldsymbol{a}|\boldsymbol{s})}{\pi_{F_{i,\theta_{\mathrm{old}}}}(\boldsymbol{a}|\boldsymbol{s})} \left(A^{F_i}(\boldsymbol{s}, \boldsymbol{a}) - \lambda_r \log \frac{\pi_{F_{i,\theta}}(\boldsymbol{a}|\boldsymbol{s})}{\pi_L(\boldsymbol{a}|\boldsymbol{s})}\right)\right]. \tag{26}$$

In this case, the objective function is computed as the expectation with respect to the follower's on-policy samples. Here, We use separate temperature parameters $\lambda_f$ and $\lambda_r$ for the forward and reverse KL terms, respectively, to ensure computational stability. By minimizing both the forward and reverse KL divergences instead of a one-sided KL divergence, we can effectively utilize samples collected by both the leader and the follower.

For simplicity, we set $\lambda_r = 0$ to perform regularization solely through the forward KL term, and clipping is applied for stable update. Thus, Eq. 26 reduces to $L_{\mathrm{SAPG},F_i}(\theta)$ in Eq. 3. Consequently, the follower policy's updated objective $L_{\mathrm{CPO},F_{i,f}}(\theta)$ and $L_{\mathrm{CPO},F_{i,r}}(\theta)$ is obtained as follows:

$$L_{\mathrm{CPO},F_{i,f}}(\theta, \lambda_f) = -\mathbb{E}_{\boldsymbol{a},\boldsymbol{s}\sim\pi_{L_{\theta_{\mathrm{old}}}}} \left[\log \pi_{F_{i,\theta}}(\boldsymbol{a}|\boldsymbol{s}) \exp\left(\frac{1}{\lambda_f} A^{F_i}(\boldsymbol{s}, \boldsymbol{a})\right)\right], \tag{27}$$

$$L_{\mathrm{CPO},F_{i,r}}(\theta) = L_{\mathrm{SAPG},F_i}(\theta).$$

## A.3 PSEUDOCODE AND COMPUTATIONAL COMPLEXITY OF THE PROPOSED METHOD

In this section, we provide the pseudocode of the proposed method and discuss the computational overhead introduced by the KL constraint and the adversarial reward. Algorithm.1 illustrates the

overall procedure of CPO. For the KL constraint, the computational difference from SAPG lies in computing the follower loss $L_{F,i}(\theta, \lambda_f)$, where the KL divergence constraint must be evaluated once for each of the five followers. Consequently, the number of auto-differentiable forward–backward passes involves roughly 12 components in CPO versus 7 in SAPG, 7 components in SAPG (five follower on-policy updates, one leader on-policy update, and one leader off-policy update) plus five additional components in CPO for follower's update from leader's samples. When the adversarial reward is used, an additional six components are introduced to train the discriminator.

Nevertheless, since data collection in SAPG typically requires about twice as much time as the training updates when using $N = 24,576$ environments, the overall wall-clock increase in training time for CPO is modest, amounting to only about 24% with the KL constraint alone and 52% when including the adversarial reward. Considering that the proposed method reaches the final performance of SAPG with less than half the number of environment steps, this overhead is acceptable.

---

**Algorithm 1:** Coupled Policy Optimization (CPO)

**Input:** Number of environments $N$, number of agents $M$, KL coefficient $\lambda_f$, adversarial weight $\lambda_{\text{adv}}$
**Output:** Updated policy parameters $\theta$, value parameters $\psi$, discriminator parameters $\xi$
**Initialize** shared policy parameters $\theta$; policy embeddings $\phi_0, \ldots, \phi_{M-1}$
**Initialize** value network $V_\psi$
**Initialize** discriminator $D_\xi$
**Initialize** $N$ environments and assign to $M$ agents (1 leader + $M-1$ followers)
**for** *each training iteration* **do**

  // -- Data collection --
  Collect trajectories $\mathcal{D}_i$ from all agents in parallel
  Compute discriminator loss $L_D(\xi)$ using $(s, a, y) \in \mathcal{D}$
  Compute adversarial reward $r_t^{\text{adv}} = \lambda_{\text{adv}} \log D_\xi(y|s_t, a_t)$
  // -- Advantage estimation --

  For each agent $i$, compute advantages $\hat{A}_t^i$ and returns $\hat{R}_t^i$ using $V_{\psi,i}$
  Sample a random follower index $j \in \{1, \ldots, M-1\}$
  Recompute leader's value/advantage on follower $j$'s data
  Recompute each follower's value/advantage on leader's data
  // -- Policy loss aggregation --
  $L_\pi \leftarrow 0$
  $L_\pi \leftarrow L_\pi + L_{L,\text{on}}(\theta)$ ;               // Leader on-policy loss
  $L_\pi \leftarrow L_\pi + L_{L,\text{off}}(\theta, j)$ ;             // Leader off-policy loss
  $L_\pi \leftarrow L_\pi + \sum_i L_{F_i,r}(\theta)$ ;          // Follower on-policy losses
  $L_\pi \leftarrow L_\pi + \beta \sum_i L_{F_i,f}(\theta, \lambda_f)$ ;     // Follower KL-regularized losses
  $L_\pi \leftarrow L_\pi + L^{\text{ent}}(\theta)$ ;             // Entropy regularization
  // -- Value loss --
  $L_V = \sum_i \|V_\psi(s_t^i) - \hat{R}_t^i\|^2$
  // -- Parameter updates --
  Update policy $\theta \leftarrow \theta - \eta_\pi \nabla_\theta L_\pi$
  Update value network $\psi \leftarrow \psi - \eta_V \nabla_\psi L_V$
  Update discriminator $\xi \leftarrow \xi - \eta_D \nabla_\xi L_D$

A.4 ABLATION STUDY ON ADVERSARIAL REWARD AND KL CONSTRAINT

In this section, we conducted an ablation study to analyze the contributions of the two key components of our method: the KL divergence constraint and the adversarial reward. We trained on two tasks, Shadow Hand and Allegro Hand, using the full **CPO** algorithm, as well as two ablated variants for analysis. The first variant, **CPO (w/o AdR)**, disables the adversarial reward by setting its scaling factor to zero ($\lambda_{\text{adv}} = 0$). The second variant, **CPO (w/o KLC)**, removes the KL divergence constraint by setting the coefficient for the solution of the forward KL minimization problem in Eq. 11 to zero ($\beta = 0$). The resulting learning curves are shown in Fig.5, while the discriminator losses are plotted in Fig.6. Additionally, the transitions of inter-policy KL divergence are visualized as color maps in Fig. 7.

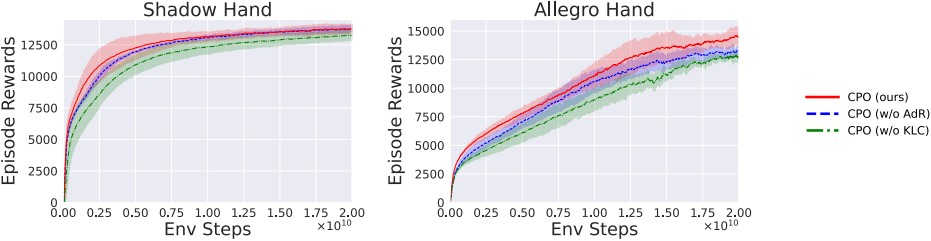

Figure 5: **Effects of KL constraint and adversarial reward on performance.** Learning curves on ShadowHand and AllegroHand tasks for three variants: full CPO (red), CPO without adversarial reward (blue), and CPO without KL constraint (green).

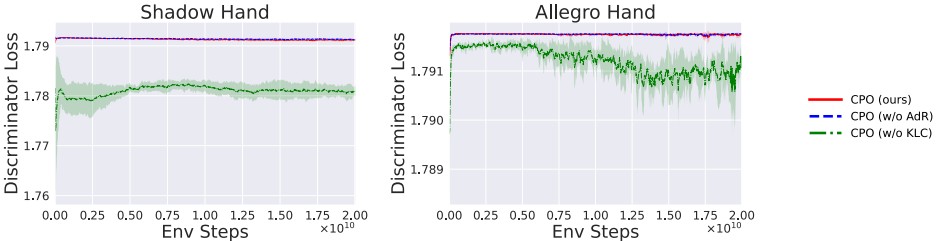

Figure 6: **Discriminator loss under different settings.** Discriminator loss during training on the ShadowHand and AllegroHand tasks for three variants: full CPO (red), CPO without adversarial reward (blue), and CPO without KL constraint (green).

As shown in Fig 5, removing the KL constraint (CPO (wo/KLC)) leads to a degradation in training performance. This suggests that, without proper regulation of policy distances, the followers explore in directions that deviate from the leader, reducing sample efficiency and training stability. This observation is further supported by the inter-policy KL divergence maps in Fig 7, where follower policies under CPO (wo/KLC) are visibly misaligned and drift far from the leader policy.

In contrast, removing the adversarial reward (CPO (w/o AdR)) results in only a marginal difference in training performance compared to the full CPO, although it tends to reduce the variance introduced by the adversarial reward across random seeds. As shown in Fig 6, the discriminator loss converges to the upper bound of random classification ($\ln 6 \approx 1.792$), indicating difficulty in distinguishing the policies regardless of the adversarial reward. In preliminary experiments, increasing the scaling factor of the adversarial reward $\lambda_{\text{adv}}$ without performance tuning made the discriminator easily distinguish between policies. In the current experiment, however, we tuned $\lambda_{\text{adv}}$ for optimal performance. As shown in Fig. 5 and Fig. 7, this results in follower policies remaining near the leader, suggesting that such alignment promotes stable and efficient learning.

Interestingly, Fig 7 shows that even without the adversarial reward, each follower's closest policy, in terms of KL divergence, is consistently the leader. This implies that the intended role of the adversarial reward, preventing overconcentration of followers, was already achieved through the KL constraint and entropy regularization alone. The performance improvement observed with the

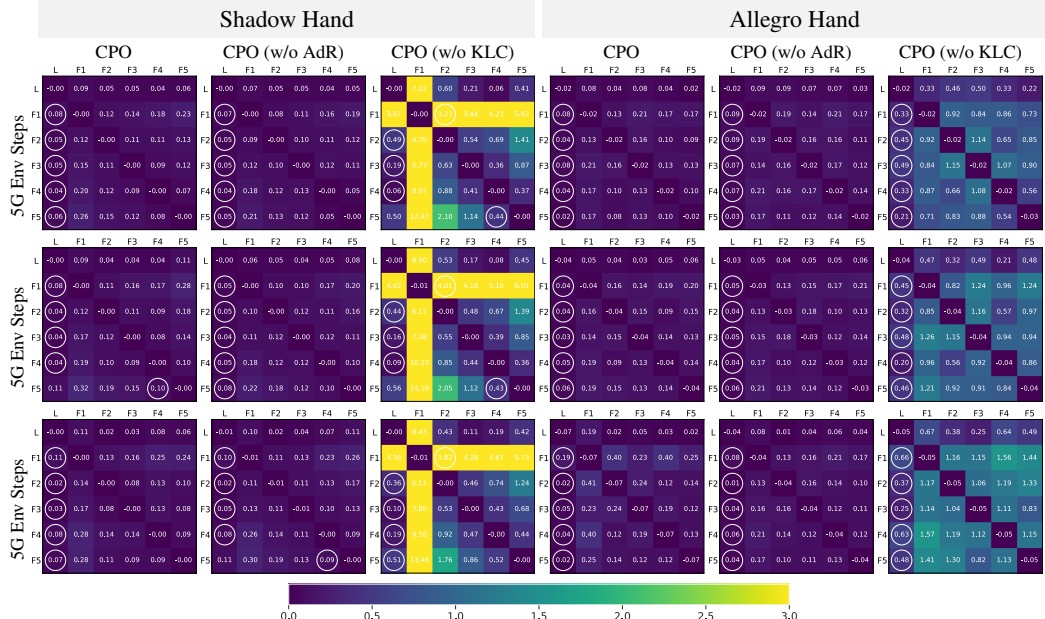

Figure 7: **Comparison of the transition of KL divergence between agents with different settings.** Each heatmap shows the KL divergence between the leader and follower policies during training. Row $i$, column $j$ indicates the forward KL from agent $i$ to agent $j$. The white circle marks the agent closest from each follower, excluding itself.

adversarial reward may stem from the uniform penalty it imposes, which encourages optimistic behaviors in the policies due to the relatively high estimated value of unexplored states. The actual impact of this regularization appears to vary depending on the task.

## A.5 TRANSITION OF INTER-POLICY KL DIVERGENCE AT HIGHER TIME-RESOLUTION

Visualizations of the transition of inter-policy KL divergence across environment steps during training are available on our project page: `https://naoki04.github.io/paper-cpo/`.

## A.6 TRAINING ENVIRONMENTS AND HYPERPARAMETERS

This section provides details on the experimental environments, task description and training hyper-parameters.

### A.6.1 EXPERIMENTAL ENVIRONMENTS

We conduct our experiments using an internal GPU cluster and a large-scale academic computing facility equipped with NVIDIA A100 GPUs. Due to differences in network environments and CPU configurations, it is difficult to make a fair comparison of training time across tasks and algorithms. However, each condition is trained for approximately one to four days to train through 20G environment steps.

### A.6.2 TASK DESCRIPTION

**Dexterous Manipulation Tasks:** For relatively simple dexterous manipulation tasks, we adopted in-hand reorientation with two types of multi-fingered hands: *ShadowHand* (24 DoF) (Andrychowicz et al., 2020) and *AllegroHand* (16 DoF). The observation space consists of joint positions and velocities, as well as object orientation and angular velocity. We used an MLP-based policy network for these tasks and set the horizon length to 8.

For more complex dexterous manipulation tasks, we adopted the *Regrasping*, *Reorientation*, and *Throw* tasks in the *Allegro-Kuka* environment (Aleksei Petrenko, 2023). In these tasks, an Allegro Hand (16 DoF) is mounted on the end of a Kuka Arm (7 DoF). To further evaluate multi-arm dexterity, we also included the *Two-Arms Reorientation* task, where two Allegro-Kuka systems simultaneously manipulate a single object in a coordinated manner. For these hand-arm manipulation tasks, we employed a policy network with a single-layer LSTM and set the horizon length to 16.

**Gripper-based Manipulation Tasks:** As non-dexterous manipulation tasks, we adopted two benchmarks: *FrankaCubePush* and *FrankaCubeStack* (8 DoF). For these tasks, we used an MLP-based policy network and set the horizon length to 8.

**Locomotion Tasks:** We adopted two locomotion benchmarks on flat ground: *Humanoid* (21 DoF) and *Anymal* (12 DoF). Although they involve high-dimensional control, their contact dynamics are relatively simpler compared to dexterous manipulation tasks, making them easier benchmarks in this context. For these tasks, we used a policy network with a single-layer LSTM and set the horizon length to 16.

### A.6.3 Training Hyperparameters

**Dexterous Manipulation Tasks:** For hand-only dexterous manipulation tasks, specifically ShadowHand and AllegroHand, we use an MLP-based Gaussian policy with an ELU activation applied after each layer. The discriminator for the adversarial reward is also implemented as an MLP with ELU activations, consisting of four hidden layers with sizes [1024, 1024, 512, 512], and is trained using a fixed learning rate equal to the initial value used for the policy. The hyperparameter settings for each task are summarized in Table 3.

For hand-arm tasks, specifically AllegroKuka Regrasping, Reorientation, Throw and Two-Arms Reorientation, we use a Gaussian policy that consists of an LSTM layer followed by an MLP with ELU activations applied after each layer. The discriminator for the adversarial reward is also implemented as an MLP with ELU activations, consisting of four hidden layers with sizes [1024, 1024, 512, 512], and is trained using a fixed learning rate equal to the initial value used for the policy. The hyperparameter settings for each task are summarized in Table 4.

**Gripper-based Manipulation Tasks:** For gripper-based manipulation tasks, specifically FrankaCubePush and FrankaCubeStack, we use a Gaussian policy that consists of an LSTM layer followed by an MLP with ELU activations applied after each layer. The discriminator for the adversarial reward is also implemented as an MLP with ELU activations, consisting of three hidden layers with sizes [256, 128, 64], and is trained using a fixed learning rate equal to the initial value used for the policy. The hyperparameter settings for each task are summarized in Table 5.

**Locomotion Tasks:** For locomotion tasks, specifically Humanoid and Anymal, we use an MLP-based Gaussian policy with an ELU activation applied after each layer. The discriminator for the adversarial reward is also implemented as an MLP with ELU activations, consisting of three hidden layers with sizes [768, 512, 256], and is trained using a fixed learning rate equal to the initial value used for the policy. The hyperparameter settings for each task are summarized in Table 6.

Table 3: **Training hyperparameters for Shadow Hand and Allegro Hand.** The upper section lists hyperparameters shared by SAPG and CPO, while the lower section lists those specific to CPO.

| Hyperparameter | Shadow Hand | Allegro Hand |
|---|---|---|
| **Common Hyperparameters (SAPG and CPO)** | | |
| Discount factor, $\gamma$ | 0.99 | 0.99 |
| GAE smoothing factor, $\tau$ | 0.95 | 0.95 |
| MLP hidden layers | [512, 512, 256, 128] | [512, 256, 128] |
| Learning rate | 5e-4 | 5e-4 |
| KL threshold for LR update | 0.016 | 0.016 |
| Grad norm | 1.0 | 1.0 |
| Entropy coefficient | 0.005 | 0 |
| Clipping factor, $\epsilon$ | 0.2 | 0.2 |
| Mini-batch size | $4 \times$ num_envs | $4 \times$ num_envs |
| Critic coefficient, $\lambda'$ | 4.0 | 4.0 |
| Horizon length | 8 | 8 |
| Bounds loss coefficient | 0.0001 | 0.0001 |
| Mini epochs | 5 | 5 |
| **CPO-Specific Hyperparameters** | | |
| $\beta$ in Eq. 11 | 0.001 | 0.0005 |
| Forward KL constraint temperature, $\lambda_f$ | 0.2 | 0.1 |
| Reverse KL constraint temperature, $\lambda_r$ | 0 | 0 |
| Adversarial reward scaling factor, $\lambda_{\text{adv}}$ | 0.01 | 0.001 |

Table 4: **Training hyperparameters for hand-arm dexterous manipulation tasks: AllegroKuka Regrasping, Reorientation, Throw and Two-Arms Reorientation.** The upper section lists hyperparameters shared by SAPG and CPO, while the lower section lists those specific to CPO.

| Hyperparameter | Regrasping | Reorientation / Two-Arms Reorientation | Throw |
|---|---|---|---|
| **Common Hyperparameters (SAPG and CPO)** | | | |
| Discount factor, $\gamma$ | 0.99 | 0.99 | 0.99 |
| GAE smoothing factor, $\tau$ | 0.95 | 0.95 | 0.95 |
| LSTM hidden units | 768 | 768 | 768 |
| MLP hidden layers | [768, 512, 256] | [768, 512, 256] | [768, 512, 256] |
| Learning rate | 1e-4 | 1e-4 | 1e-4 |
| KL threshold for LR update | 0.016 | 0.016 | 0.016 |
| Grad norm | 1.0 | 1.0 | 1.0 |
| Entropy coefficient | 0 | 0.005 | 0 |
| Clipping factor, $\epsilon$ | 0.1 | 0.1 | 0.1 |
| Mini-batch size | $4 \times$ num_envs | $4 \times$ num_envs | $4 \times$ num_envs |
| Critic coefficient, $\lambda'$ | 4.0 | 4.0 | 4.0 |
| Horizon length | 16 | 16 | 16 |
| LSTM Sequence length | 16 | 16 | 16 |
| Bounds loss coefficient | 0.0001 | 0.0001 | 0.0001 |
| Mini epochs | 2 | 2 | 2 |
| **CPO-Specific Hyperparameters** | | | |
| $\beta$ in Eq. 11 | 0.001 | 0.001 | 0.001 |
| Forward KL constraint temperature, $\lambda_f$ | 0.2 | 0.2 | 0.1 |
| Reverse KL constraint temperature, $\lambda_r$ | 0 | 0 | 0 |
| Adversarial reward scaling factor, $\lambda_{\text{adv}}$ | 0 | 0 | 0 |

Table 5: **Training hyperparameters for gripper-based manipulation tasks: FrankaCubePush and FrankaCubeStack.** The upper section lists hyperparameters shared by SAPG and CPO, while the lower section lists those specific to CPO.

| Hyperparameter | FrankaCubePush | FrankaCubeStack |
|---|---|---|
| **Common Hyperparameters (SAPG and CPO)** | | |
| Discount factor, $\gamma$ | 0.99 | 0.99 |
| GAE smoothing factor, $\tau$ | 0.95 | 0.95 |
| LSTM hidden units | 256 | 256 |
| MLP hidden layers | [256, 128, 64] | [256, 128, 64] |
| Learning rate | 5e-4 | 5e-4 |
| KL threshold for LR update | 0.008 | 0.008 |
| Grad norm | 1.0 | 1.0 |
| Entropy coefficient | 0.005 | 0.005 |
| Clipping factor, $\epsilon$ | 0.2 | 0.2 |
| Mini-batch size | $4 \times$ num_envs | $4 \times$ num_envs |
| Critic coefficient, $\lambda'$ | 4.0 | 4.0 |
| Horizon length | 16 | 16 |
| LSTM Sequence length | 16 | 16 |
| Bounds loss coefficient | 0.0001 | 0.0001 |
| Mini epochs | 8 | 8 |
| **CPO-Specific Hyperparameters** | | |
| $\beta$ in Eq. 11 | 0.0005 | 0.0005 |
| Forward KL constraint temperature, $\lambda_f$ | 0.1 | 0.1 |
| Reverse KL constraint temperature, $\lambda_r$ | 0 | 0 |
| Adversarial reward scaling factor, $\lambda_{\text{adv}}$ | 0 | 0 |

Table 6: **Training hyperparameters for locomotion tasks: Humanoid and Anymal.** The upper section lists hyperparameters shared by SAPG and CPO, while the lower section lists those specific to CPO.

| Hyperparameter | Humanoid | Anymal |
|---|---|---|
| **Common Hyperparameters (SAPG and CPO)** | | |
| Discount factor, $\gamma$ | 0.99 | 0.99 |
| GAE smoothing factor, $\tau$ | 0.95 | 0.95 |
| MLP hidden layers | [768, 512, 256] | [768, 512, 256] |
| Learning rate | 5e-4 | 3e-4 |
| KL threshold for LR update | 0.008 | 0.008 |
| Grad norm | 1.0 | 1.0 |
| Entropy coefficient | 0.002 | 0.002 |
| Clipping factor, $\epsilon$ | 0.2 | 0.2 |
| Mini-batch size | $4 \times$ num_envs | $4 \times$ num_envs |
| Critic coefficient, $\lambda'$ | 4.0 | 4.0 |
| Horizon length | 8 | 8 |
| Bounds loss coefficient | 0.0001 | 0.0001 |
| Mini epochs | 5 | 5 |
| **CPO-Specific Hyperparameters** | | |
| $\beta$ in Eq. 11 | 0.001 | 0.001 |
| Forward KL constraint temperature, $\lambda_f$ | 0.2 | 0.2 |
| Reverse KL constraint temperature, $\lambda_r$ | 0 | 0 |
| Adversarial reward scaling factor, $\lambda_{\text{adv}}$ | 0 | 0 |

### A.7 ENTROPY REGULARIZATION ABLATION ON SAPG

In this section, we investigate the relationship between follower-policy misalignment in SAPG and the entropy regularization term. To this end, we conducted an ablation study on the entropy coefficient across three tasks: Shadow Hand, AllegroKuka Regrasping, and AllegroKuka Reorientation. The learning curves are shown in Fig. 8, and the inter-policy KL divergence during training is visualized in Fig. 9.

From Fig. 8, consistent with observations in the SAPG paper (Singla et al., 2024), the effect of entropy regularization on training performance varies significantly across tasks. More importantly, Fig. 9 further shows that introducing entropy regularization in SAPG consistently increases inter-policy KL divergence and often leads to severe misalignment.

These results suggest that, while entropy regularization indeed promotes exploration and yields diverse data, it also induces policy misalignment that destabilizes the leader's learning, supporting our main claim. In contrast, CPO suppresses this disadvantage while still promoting exploration within a KL-bounded region, enabling stable leader updates and achieving superior performance.

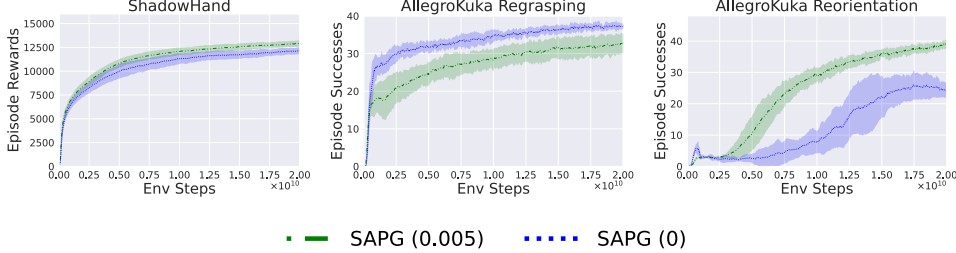

Figure 8: **Comparison of SAPG with and without entropy regularization.** The values in parentheses indicates the entropy coefficients. The effect of entropy regularization on learning performance varies across tasks.

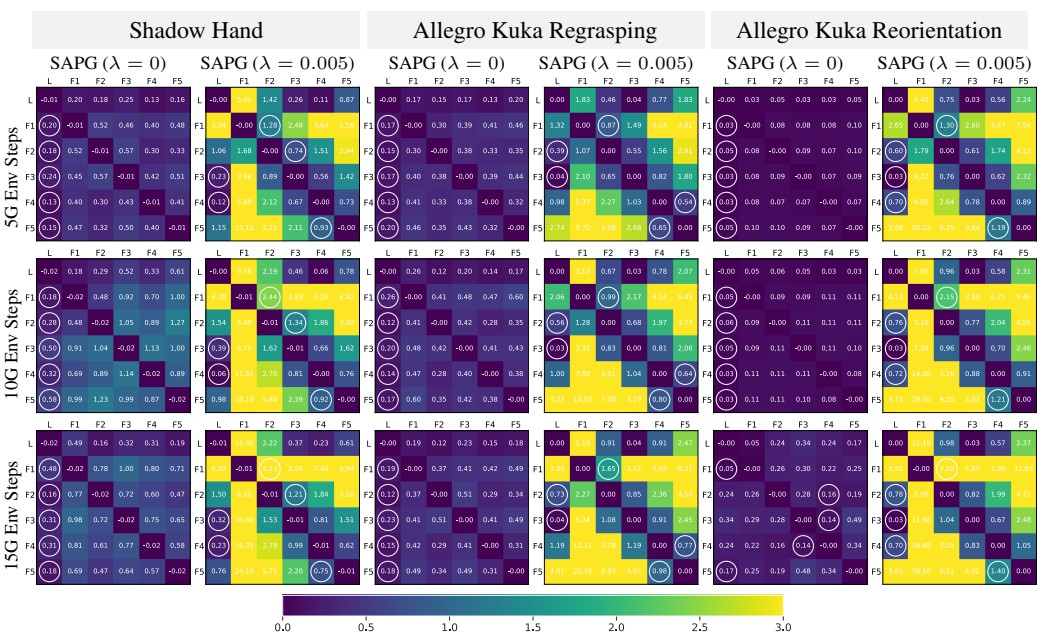

Figure 9: **Comparison of the transition of KL divergence in SAPG with and without entropy regularization.** Each heatmap shows the KL divergence between the leader and follower policies during training. Row $i$, column $j$ indicates the forward KL from agent $i$ to agent $j$. The white circle marks the agent closest from each follower, excluding itself. It is demonstrated that entropy regularization causes follower's misalignment from the leader policy.

