# OpenReview forum: "Rethinking Policy Diversity in Ensemble Policy Gradient in Large-Scale Reinforcement Learning"
_ICLR.cc/2026/Conference — ICLR 2026 Poster_

### Official Review · Reviewer_ZcZ8 · 2025-10-29

**Soundness:** 3
**Presentation:** 3
**Contribution:** 2
**Rating:** 4
**Confidence:** 5

**Summary:**

This paper shows that in large-scale reinforcement learning, blindly pursuing policy diversity across multiple policies reduces sample efficiency and training stability. To address this, it proposes Coupled Policy Optimization (CPO): within a leader–follower framework it regulates diversity via a KL constraint and an adversarial intrinsic reward, and it uses importance sampling to efficiently absorb data from followers. Experiments show that CPO achieves higher sample efficiency and more stable, superior final performance than existing baselines on a range of dexterous manipulation tasks.

**Strengths:**

1. This paper proposes CPO, which within a leader-follower framework regulates the exploration radius and coverage via a KL constraint to the leader policy coupled with an adversarial intrinsic reward, while retaining efficient use of off-policy data.
2. This paper uses upper-bound analysis to connect the leader-follower KL with IS ratio deviation, ESS, and gradient bias, and, drawing on Pinsker-type inequalities and PPO clipping analysis, offers a quantitative justification for why distance control is needed.
3. On dexterous manipulation tasks under ultra-large parallel settings, CPO shows higher sample efficiency and more stable, superior final performance, supported by ablations and visualizations.

**Weaknesses:**

1. CPO essentially adds a KL constraint to the leader and an adversarial intrinsic reward on top of a leader-follower framework. The novelty is incremental.
2. The discriminator and KL regularization introduce additional compute overhead and hyperparameters, yet the paper lacks wall-clock comparisons under equal compute and equal interaction budgets, as well as a systematic hyperparameter sensitivity analysis.
3. The experimental evidence focuses mainly on dexterous manipulation with a single platform/parallelism setting, and the breadth of tasks and difficulty axes is insufficient to support conclusions.

**Questions:**

See Weaknesses above.

---

> ### Author Response · Authors · 2025-11-20
> **Response to Reviewer ZcZ8**
>
> Thank you for the valuable feedback.
>
> Below, **we address your comments regarding (1) computational overhead**, **(2) evaluation across diverse environments**, **(3) hyperparameter sensitivity**, and **(4) the contribution of our method**.
>
> ---
> # 1. Evaluation on computational overhead
> Thank you for raising this point. Due to limitations of computational resources, our experiments were conducted across multiple clusters, which makes a strictly controlled wall-clock comparison difficult. However, **we provide a complexity analysis that details the additional cost** introduced by the KL constraint. **In short, the improvement in time-to-performance more than compensates for the per-iteration cost**.
>
> **The number of auto-differentiable forward–backward passes involves roughly 12 components in CPO versus 7 in SAPG**, seven components in SAPG (five follower on-policy updates, one leader on-policy update, and one leader off-policy update) plus five additional components in CPO for follower’s update from leader’s samples. Nevertheless, **the overall wall-clock increase including simulation time for CPO is modest, amounting to only about 25%**.
>
> On the other hand, **CPO achieves over 2× faster time-to-performance to reach SAPG’s final performance** on tasks with significant misalignment. For example, in the AllegroKukaReorientation task, **SAPG required $\mathbf{20\times10^9}$ samples to reach 38.75 episode successes**, whereas **CPO achieved the same performance with only $\mathbf{7.5\times10^9}$ samples**. Importantly, **CPO also surpasses SAPG in final performance** on multiple tasks.
>
> Please refer to **Appendix A.3** for details.
>
> ---
> # 2. Additinal experiments on non-dexterous envirionments.
> **To address your request, we conducted additional experiments on two non-dextrous manipulation tasks**, i.e. FrankaCubePush and FrankaCubeStack, to further evaluate the generalizability of our method.
>
> In both tasks, **CPO achieved superior final performance compared to all baselines, demonstrating the generalizability**. The results have been added to **Fig. 5 in Appendix A.5 of the revised manuscript**, together with the previously reported locomotion experiments.
>
> Thanks to your suggestion, we think that these results further strengthen the advantage of the proposed method.
>
> ---
> # 3. Systematic hyperparameter sensitivity analysis
> We appreciate the comment regarding hyperparameter sensitivity. In the paper, **we provide ablations on (1) the strength of the KL constraint λ, and (2) the presence/absence of the KL constraint and the adversarial reward**.
>
> **(1)** The ablations on the strength of KL constraint show that **CPO maintains superior performance to SAPG across a wide range of λ** values. (See **Section 7.2**)
>
> **Regarding practical guidance for choosing $\lambda$, we recommend starting from a large value (e.g., $\lambda$ = 0.5)** and monitoring the inter-policy KL divergence and the ESS. **If misalignment is observed, typically indicated by diverging KL distances or rapidly decreasing ESS, we suggest decreasing $\lambda$ (e.g., to 0.2 or 0.1)**.
>
> **(2)** The ablation study on the two components show that the **KL constraint is the primary driver of performance gains, while the adversarial reward provides a small but consistent benefit** by preventing follower collapse during training. (See Appendix A.5)
>
> We appreciate your constructive feedback and would be happy to consider any further suggestions you may have.
>
> ---
> # 4. Contribution of the paper
> While the KL constraint and adversarial reward may seem simple, **our contributions go beyond the method itself**. This paper also provides:
>
> 1.  **A new theoretical perspective** showing that **the expected IS-ratio deviation is the key factor** governing stability and sample efficiency in leader's off-policy updates, and that it is **upper-bounded by the forward KL**. **This connection has not been analyzed in prior ensemble-RL work.**
>
> 2. **Empirical results that align closely with the theory**, demonstrating that the KL constraint effectively suppresses IS-ratio deviation, increases ESS, and leads to consistent improvements in sample efficiency and stability across tasks.
>
> 3. **Comprehensive inter-policy KL analyses** showing that uncontrolled diversity causes severe misalignment in SAPG, whereas CPO induces a stable, well-structured follower distribution around the leader.
>
> We believe that **our theoretical findings are significant and will contribute to advancing algorithms for massively parallel environments**. Although the resulting technique is simple, **our theoretically grounded method demonstrates substantial performance improvement**.
>
> ---
> Thank you again for your insightful comments. We hope our addtional experiments and responses address your concerns and clarify the contributions of our work.

---

> > ### Comment · Reviewer_ZcZ8 · 2025-11-26
> >
> > Thank you for the authors’ response. The additional experiments and explanations address some of my concerns (Weaknesses 2 and 3). I will raise my score to 6.

---

> > > ### Author Response · Authors · 2025-11-26
> > >
> > > We appreciate your careful reassessment and your updated score. Your constructive comments were invaluable in improving the manuscript. Thank you again for your time and support.

---

### Official Review · Reviewer_nLNK · 2025-10-30

**Soundness:** 3
**Presentation:** 3
**Contribution:** 2
**Rating:** 8
**Confidence:** 4

**Summary:**

In this work, the authors theoretically analyze the impact of inter-policy diversity on learning efficiency in policy ensembles, and propose Coupled Policy Optimization (CPO), a method that regulates diversity via KL constraints between policies. The proposed CPO enables effective exploration and outperforms strong baselines (including SAPG, PBT, and PPO) across multiple dexterous manipulation tasks, showing advantages in both sample efficiency and final performance. Additionally, the experimental results are robust, and the paper is fluently written.

**Strengths:**

1. The paper is well-structured and clearly written, ensuring good readability.

2. The authors rethink the existing SAPG method by providing theoretical analysis and key insights, which effectively motivate the design of the proposed CPO.

3. The experimental results are robust and supported by sufficient evidence, and the overall writing flow is smooth.

**Weaknesses:**

1. More concrete examples could be added to illustrate and validate the key theoretical insights, which would strengthen the persuasiveness of the work.

2. The design of the proposed CPO is relatively straightforward, as it only uses KL divergence to constrain the distance between follower and leader policies, lacking further optimization or innovative adjustments.

- 2.1 The selection of the lambda hyperparameter in practice is somewhat heuristic, with no clear justification provided for its choice.

- 2.2 KL divergence is a relatively trivial and commonly used distance metric. In the field of policy diversity research, numerous works have proposed alternative diversity metrics [1-3]. A more in-depth discussion of these existing metrics, along with the rationale for selecting KL divergence in this work, would improve the justification for the method’s design.

[1] Effective diversity in population based reinforcement learning.

[2] Policy space diversity for non-transitive games.

[3] Quality-similar diversity via population based reinforcement learning.

**Questions:**

1. Are there any specific guidelines or recommendations for the selection of the lambda hyperparameter?

2. The paper adopts SAPG as its baseline method. Could the authors explain why SAPG outperforms Population-Based Training (at least in their experimental settings)?

3. Why was KL divergence chosen as the policy distance metric? Would using reverse KL divergence alter the conclusions of this work?

4. Why is the score of SAPG very low in Two-Arms Reorientation in Table 1, but the proposed method, which is based on SAPG, achieves the highest score?

---

> ### Author Response · Authors · 2025-11-20
> **Response to Reviewer nLNK**
>
> Thank you for your constructive feedback. Our work analyzes the leader’s off-policy update dynamics in a leader–follower architecture and, based on this analysis, introduces KL regularization and an auxiliary adversarial reward to improve training stability and sample efficiency.
>
> Below, we address your concerns regarding **(1) the choice of KL divergence**, **(2) additional experiments**, **(3) the selection of $\lambda$**, **(4) why SAPG outperforms PBT in our setting**, and **(5) the performance gap in Two-Arms Reorientation**.
>
> ---
> # 1-1. Why KL divergence?
> Thank you for the insightful question. In our work, **we adopted the forward KL divergence (from follower to leader) based on the theoretical analysis**.
>
> In Section 4, we show that **the deviation of the IS ratio from 1 is the key quantity affecting the leader’s off-policy update**. When this deviation is large, off-policy learning becomes unstable.
> More concretely:
> - **large deviation → low Effective Sample Size (Prop. 1)**,
> - **large deviation → large PPO clipping bias (Prop. 2)**.
>
> Crucially, **Proposition 3** proves that **this IS-ratio deviation is upper-bounded by the forward KL divergence between the follower and the leader policies**.
> Therefore, constraining the forward KL directly controls the source of instability in off-policy updates, providing the theoretical foundation for the performance improvements observed in our method.
>
> ---
> # 1-2. Could inverse KL be used?
> In principle, **any divergence measure that upper-bounds the IS-ratio deviation (or its KL upper bound) could serve as a valid alternative**. Designing such metrics is an interesting direction for future work.
>
> However, **using the reverse KL is challenging because it is unclear whether it can upper-bound the IS-ratio deviation**; the deviation is estimated from follower samples, whereas the reverse KL is evaluated under the leader distribution.
>
> ---
> # 1-3. Regarding the suggested alternative diversity metrics
>
> Determinantal Diversity [1], sequence-form Bregman divergences [2], and QSD [3] are indeed interesting prior works. **However, applying them in our setting also require theoretical guarantee on off-policy update stability, which is essential in our framework**, as described above.
> Also, sequence-form Bregman divergences assume finite action spaces; in continuous control settings, the realization plan becomes infinite-dimensional, making [2] non-trivial to apply.
>
> We agree that these works offer valuable context. **We have added the corresponding citations to Section 2.3 (Related Work) in the revised manuscript**.
>
> ---
> # 2. Additional experiments
> **We conducted additional experiments on two non-dextrous manipulation tasks**, i.e. FrankaCubePush and FrankaCubeStack, to further evaluate the generalizability of our method.
> In both tasks, **CPO achieved superior final performance compared to all baselines, demonstrating the generalizability**. The results have been added to **Fig. 5 in Appendix A.5 of the revised manuscript**, together with the previously reported locomotion experiments.
>
> ---
> # 3. How to choose $\lambda$?
> When $\lambda$ is large, the KL regularization between the leader and follower policies becomes weaker. As discussed in Section 7.2, **we recommend starting from a large value (e.g., $\lambda$ = 0.5)** and monitoring the inter-policy KL divergence and the ESS. **If misalignment is observed, typically indicated by diverging KL distances or rapidly decreasing ESS, we suggest decreasing $\lambda$ (e.g., to 0.2 or 0.1)**.
>
> ---
> # 4. Why does SAPG outperform PBT?
> **While PBT encourages broad exploration** through hyperparameter diversity, **it does not reuse data across policies or generations**, which limits its learning ability on high-dimensional or contact-rich tasks that require both broad and deep exploration.
>
> In contrast, **SAPG aggregates all trajectories collected by follower policies to update the leader**, enabling both **wide and deep exploration**. As a result, SAPG achieves much better sample efficiency and final performance in large-scale dexterous manipulation tasks.
>
> However, for tasks that require online adaptation of the hyperparameters or do not require deep exploration, PBT may outperform SAPG. This behavior is also consistent with our results in **Section A.4 (Fig. 5)**: **in simpler locomotion tasks** such as Anymal-Walk (Flat) and Humanoid-Walk, **SAPG performs similarly to, or slightly worse than, PBT**, further highlighting the advantage and disadvantage of the two methods.

---

> > ### Author Response · Authors · 2025-11-20
> > **Response to Reviewer nLNK**
> >
> > # 5. Why does CPO succeed on Two-Arms Reorientation while SAPG fails?
> >
> > This question is closely tied to **when misalignment arises in SAPG**. **In Appendix A.8, we conducted an additional ablation study on entropy-regularization term in SAPG and reported that the strong entropy-regularization leads to the misalignment issue in SAPG**, resulting inferior performance compared to CPO. In Two-Arms Reorientation, SAPG was severely suffered from the misalignment, whereas CPO resolves it and achieves significantly higher performance.
> >
> > (**See the response 1 to Reviewer FSNQ and Section A.8 for supporting ablations.**)
> >
> > ---
> > Thank you again for your insightful comments. We hope our renpose and addtional experiments satisfactorily address your questions.

---

> > > ### Comment · Reviewer_nLNK · 2025-11-27
> > > **Thank you for your response!**
> > >
> > > Thanks for addressing my question. The question regarding KL divergence and the experiment is well addressed. And I  maintain my score and vote for acceptance.

---

> > > > ### Author Response · Authors · 2025-11-27
> > > >
> > > > Thank you for reviewing our response. Your comments regarding divergence metrics were very helpful in improving the manuscript. We sincerely appreciate your time and effort.

---

### Official Review · Reviewer_FSNQ · 2025-10-30

**Soundness:** 3
**Presentation:** 3
**Contribution:** 2
**Rating:** 4
**Confidence:** 3

**Summary:**

This paper addresses the problem of excessive policy diversity in ensemble policy gradient methods  by improving the policy optimization method for followers. Specifically, it introduces KL constraints during the follower updates to regulate the distance to the leader, and incorporates an adversarial reward to prevent policy overconcentration . The proposed method (CPO) achieves structured and efficient exploration. Experiments demonstrate that this method can effectively improve sample efficiency and final performance.

**Strengths:**

1. The paper's insight is presented very clearly, and the logic is rigorous; we can easily follow the author's train of thought and logic to understand the method. The paper's writing quality is high.

2. The theoretical derivations are thorough, and the experimental validation is comprehensive. The visualization of KL divergence changes in Figure 4 is very valuable.

3. The method achieves a significant breakthrough on a high-difficulty task (Two-Arms Reorientation).

**Weaknesses:**

1. The model's generalizability appears limited. It is only effective in specific environments, such as AllegroHand, where follower policies are prone to significant divergence (resulting in high variance). In contrast, on other tasks like Regrasping, the performance improvement is not as pronounced .

2. The training cost is somewhat high. The paper's CPO method requires more backpropagation components (roughly 12 vs. 7 for SAPG) and more wall-clock training time per iteration (approximately 25% more) .

3. Although the design of the Adversarial Reward is interesting—leveraging the idea from DIAYN to classify policies based on state-action pairs —the ablation study shows its impact on the results is not significant. The necessity of this module is therefore questionable .

**Questions:**

1. I wonder whether "misaligned significantly" phenomenon consistently occur in all "complex tasks". Could you provide more analysis on what types or characteristics of environments are prone to causing this "misaligned significantly" state? Can this be primarily attributed to the environment's exploration complexity? Could I regard the KL constraint (CPO's core mechanism) essentially act as a mechanism to reduce exploration and increase exploitation;

2. Are the marginal benefits of improving the IS and ESS metrics on the final performance limited? I observed that the ESS was optimized by over 40-fold (e.g., from 2.23% to 94.1% in ShadowHand ), yet the final sample efficiency (as shown in the reward curves ) only showed a limited improvement (approx. 2-2.5x).

---

> ### Author Response · Authors · 2025-11-20
> **Response to Reviewer FSNQ**
>
> We sincerely appreciate you for the careful reading of our paper and the insightful comments. As you correctly understood, our work builds on a theoretical analysis of the leader’s off-policy learning dynamics and introduces KL constraints, together with an auxiliary adversarial reward, to stabilize training and improve sample efficiency.
>
> Below, we address your concerns regarding **the emergence of misalignment, generalizability to non-dextrous tasks, the role of ESS, and the computational cost of our method**.
>
> ---
> # 1. When does misalignment occur?
> Thank you for raising this important question. Misalignment can arise from various factors, including the dimensionality of the action space and task-specific characteristics. **Regarding SAPG, we think that misalignment often occurs when SAPG is combined with strong entropy regularization in tasks that require extensive exploration**.
>
> In SAPG, an entropy-maximization term (defined below) is added to the policy objective to encourage exploration, and its weight is tuned for each task. In our initial submission, we used the same hyperparameters as the original SAPG implementation.
>
> \begin{equation}
>     L ^{\mathrm{ent}}(\theta ,j) = \mathbb{E} _{s \sim {\pi _L} _{ \theta _{ \mathrm{old}}}, { \pi _F} _{j, \theta _{ \mathrm{old}}}} [ \mathcal{H} [ \pi _{L _{\theta}}( \cdot | s )]] + \sum _{i \in \lbrace 0, \dots ,M-2 \rbrace} \mathbb{E} _ {s \sim {\pi} _  {F _{i, \theta _{\mathrm{old}}}}} [ \mathcal{H} [ \pi _ {F _ {i, \theta }} (\cdot | s )]],
> \end{equation}
> where $i$ indicates follower indices and $j$ indicates a randomly sampled follower index for leader's off-policy update.
>
> **To address the request from Reviewer FNSQ, we conducted an additional ablation study on the entropy coefficient in SAPG** and added the resulting learning curves and policy KL visualizations to **Section A.8** of the revised manuscript.
>
> From **Fig. 9**, consistent with observations in the SAPG paper, the effect of entropy regularization on training performance varies significantly across tasks. More importantly, **Fig. 10** further shows that **introducing entropy regularization in SAPG consistently increases inter-policy KL divergence** and often leads to severe misalignment.
>
> These results suggest that, **while entropy regularization indeed promotes exploration and yields better sample efficiency as shown in Fig. 9, it also induces policy misalignment that destabilizes the leader’s learning**, supporting our main claim.
> In contrast, **CPO suppresses this disadvantage while still promoting exploration within a KL-bounded region**, enabling stable leader updates and achieving superior performance.
>
> ---
> # 2. Generalizability to non-dexterous tasks
> **To address your concern, we conducted additional experiments on two non-dextrous manipulation tasks**, i.e. FrankaCubePush and FrankaCubeStack, to further evaluate the generalizability of our method.
>
> In both tasks, **CPO achieved superior final performance compared to all baselines, demonstrating the generalizability**. The results have been added to **Fig. 5 in Appendix A.5 of the revised manuscript**, together with the previously reported locomotion experiments.
>
> ---
> # 3. Are the marginal benefits of improving the IS and ESS metrics on the final performance limited?
>
> ESS evaluates the degree of weight concentration introduced by importance sampling, defined as follows.
>
> \begin{equation}
> ESS = \frac{1}{ \sum _{i=1} ^N \tilde{w} _ i^2},
> \quad \tilde{w} _i = \frac{w _i} {\sum _{j=1} ^N w _j},
> \end{equation}
> where $w_i$ is the IS ratio.
>
> **An ESS close to 1 indicates that samples contribute evenly to learning, whereas an ESS near 0 means that only a small subset of samples dominates the gradient estimate while most samples are effectively discarded.**
>
> Importantly, **ESS is not linearly related to the agent’s expected return**, so a 40× improvement in ESS does not imply a 40× acceleration in learning. Instead, **higher ESS primarily stabilizes gradient estimates and reduces variance in leader updates**, which in turn contributes to more reliable and sample-efficient learning, as also reflected in our experimental results.
>
> ---
> # 4. Computational efficiency
> We appreciate your careful examination of the computational cost. As you noted, **while CPO introduces a modest per-iteration overhead (~25%), it achieves over 2× faster time-to-performance to reach SAPG’s final performance** on tasks with significant misalignment. For example, in the AllegroKukaReorientation task, **SAPG required 20 G samples to reach 38.75 episode successes**, whereas **CPO achieved the same performance with only 7.5 G samples**. This makes the additional computation well justified.
>
> ---
> Thank you again for your helpful feedback. **We believe that the additional experiments and analysis further have clarified mechanism of the misalignment and performance of the proposed method.**

---

### Official Review · Reviewer_hCse · 2025-10-31

**Soundness:** 3
**Presentation:** 3
**Contribution:** 3
**Rating:** 8
**Confidence:** 3

**Summary:**

This paper tackles the challenge of scaling reinforcement learning (RL) to large numbers of parallel environments, where a single policy’s limited exploration capacity can hinder performance. The authors focus on ensemble-based policy gradient methods that use multiple policies to promote exploration but note that excessive diversity can harm learning stability and efficiency. They present Coupled Policy Optimization (CPO), a method that constrains inter-policy divergence via KL regularization to balance exploration and coordination among ensemble members. Theoretical analysis establishes how controlled diversity improves learning efficiency, and empirical evaluations on dexterous manipulation tasks show that CPO outperforms prior methods such as SAPG, PBT, and PPO in both sample efficiency and final performance. Additional analysis demonstrates that follower policies self-organize around a leader, yielding structured and effective exploration. Overall, the work highlights the importance of regulated diversity for stable and efficient learning in ensemble policy gradient frameworks.

**Strengths:**

* It appears to be a novel and difficult task to introduce exploration incentives to the follower agents at risk of destabilizing the already off-policy training, however this paper appears to utilize the KL divergence and the discriminator in a way that promotes some exploration for the follower agents without training collapsing.
 * The paper is well written, and utilizes the background and method section well. The results are well thought out.
* Although the main algorithmic contribution appears to be simple, the KL divergence, the addition is well justified and explored thoroughly in the paper.

**Weaknesses:**

* I would have liked to know why a discriminator was chosen specifically, in comparison to other exploration based algorithms, in particular there are other methods that do not require the additional external training or usage of a functional approximation [1].

* Going further on the second point, although the KL divergence effects were explained, section 5.2 appears to be the only discussion on the usage of an exploration algorithm, and it appears that there is little explanation on how the adversarial signal improves the performance of the algorithm. If this component of the paper was discussed or there was more time spent analyzing this more somewhere in the paper I would be more confident staying at an accept.

[1] Susan Amin, Maziar Gomrokchi, Harsh Satĳa, Herke van Hoof, & Doina Precup. (2021). A Survey of Exploration Methods in Reinforcement Learning.

**Questions:**

* I would say that although DIAYN uses a discriminator, their work is primarily interested in gathering skills and maximizing entropy, would it be reasonable to introduce to the reader as (ICM)[2] which explicitly relies on prediction error? I don't consider this an issue, but there are other methods that leverage prediction error that might make a better fit.

[2] Pathak, D., Agrawal, P., Efros, A.A., Darrell, T.: Curiosity-driven exploration by
self-supervised prediction. In: International Conference on Machine Learning, pp.
2778–2787 (2017). PMLR

---

> ### Author Response · Authors · 2025-11-20
> **Response to Reviewer hCse**
>
> We sincerely thank you for the thoughtful and constructive feedback. As you correctly understood, our work builds on a theoretical analysis of the leader’s off-policy learning dynamics and introduces KL constraints, together with an auxiliary adversarial reward, to stabilize training and improve sample efficiency.
>
> Below, **we address your questions regarding the adversarial reward and the choice of DIAYN**.
>
> ---
>
> # 1. Why does the adversarial reward help?
> In our method, follower policies are trained to explore within a KL-bounded neighborhood around the leader to preserve the stability of the leader’s off-policy PPO update.
>
> While the KL constraint prevents harmful misalignment from the leader, it also **indirectly pulls followers closer to one another**, which can reduce the diversity of their state–action coverage within the neighborhood.
>
> **The DIAYN-based adversarial reward**, defined as follows, **serves as an auxiliary mechanism to prevent this over-concentration by encouraging followers to produce distinguishable behaviors.**
> \begin{align}
> L_{D}(\xi) = -\mathbb{E}_{(s _t, a _t, y)\sim \mathcal{D}} [ \log D _{ \xi } ( y | s _t, a _t)  ], \quad r ^{\text{adv}} _t (s _t, a _t, y) = \lambda _{\text{adv}} \log{D _\xi (y | s _t, a _t)},
> \end{align}
>
> where $D_\xi(y|{s}_t,{a}_t)$ denotes the discriminater to predict the index $y \in \lbrace 0, \dots,M-1 \rbrace $ of the policy given a state-action pair.
>
>
> We conducted an ablation study to examine the impact of the adversarial reward and the KL constraint, as reported in **Appendix A.5**. The result indicates that **the KL constraint is the primary driver of performance gains, while the adversarial signal offers a small but consistent benefit by preventing follower collapse** during training.
>
> **We appreciate your comment and added a motivating explanation at the start of Section 5.2** in the Related Work.
>
> ---
>
> # 2. Why DIAYN instead of ICM or other prediction-error–based exploration?
>
> We appreciate your insightful suggestion. **Curiosity-based methods such as ICM and RND**[3] are designed to encourage agents to explore unseen states, **but they are not intended for scenarios where multiple policies perform rollouts in parallel**. Consequently, these methods do not explicitly account for the distance between policies executing rollouts simultaneously.
>
> **Our objective is not only to encourage exploration. Our motivation is to prevent follower policies from collapsing into overlapping state–action regions within the KL-bounded neighborhood of the leader**, which would reduce effective sample diversity.
>
> For this purpose, **we employed DIAYN, which explicitly separates the followers’ state–action distributions**. Exploring alternative mechanisms that satisfy this objective would be an interesting direction for future work.
>
> **We have updated the expression in Section 5.2 to make this design choice clear based on your feedback.**
>
> [3]Burda, Yuri, et al. "Exploration by random network distillation." arXiv preprint arXiv:1810.12894 (2018).
>
> ---
> We thank you again for your thoughtful review and hope that our responses address your concerns.

---

> > ### Comment · Reviewer_hCse · 2025-11-22
> >
> > Thank you,
> >
> > The added ablation along with the explanation provided, addresses my main concerns.
> >
> > I keep my score.

---

> > > ### Author Response · Authors · 2025-11-24
> > >
> > > Thank you for reviewing our response and for the constructive feedback. Your comments were very helpful in improving the manuscript. We sincerely appreciate your time and effort.

---

### Comment · Area_Chair_Y5Gt · 2025-11-21
**Author-Reviewer Discussion**

Dear reviewers,

Please review the authors' response and adjust your rating accordingly. If you have any further questions, please discuss with the authors further.

AC

---

### Public Comment · ~Zhengpeng_Xie1 · 2025-11-27

Hi, I think you should cite SPO [1], especially regarding the increase of bias by ratio clipping in PPO, which we believe was first noted by SPO.

[1] Z Xie et al. Simple policy optimization. ICML 2025.

---

> ### Public Comment · ~Zhengpeng_Xie1 · 2025-11-27
>
> Specifically, lines 223 to 230 following Proposition 2:
>
> _PPO ensures learning stability by clipping the IS ratio; however, this introduces bias into the gradient estimate. As the IS deviation increases, the effect of clipping becomes more pronounced, leading to greater bias and destabilizing the leader’s learning._
>
> which we believe represent an insight **first proposed** by SPO.

---

> > ### Author Response · Authors · 2025-11-28
> >
> > Thank you for pointing out the valuable prior work regarding the impact of IS ratio deviation on PPO.
> >
> > After carefully reading the SPO paper, we realized that it explicitly **establishes a connection between IS ratio deviation and a lower bound on performance improvement** in Eq. (10) in the paper.
> >
> > **In response to your feedback, we have updated the manuscript and added the appropriate citation in Section 4, lines 251–253**, as shown below:
> > > Consequently, introducing a constraint on the KL divergence between the leader and follower policies alleviates the IS ratio deviation. **Schulman et al. (2015) and Xie et al. (2025) also argue that as long as the KL divergence or IS ratio deviation between the target and behavior policies remains small, reduces update error from distribution shift, and performance improvement is guaranteed.** These motivate the need for KL-based coupling between leader and followers, to regulate policy diversity in ensemble policy gradient methods.
> >
> >
> > Regarding the theoretical aspect, we would like to clarify the following. While Xie et al. (2025) point out that PPO clipping cannot enforce trust region constraints and that IS ratio deviation can hinder monotonic policy improvement, **the paper does not provide a mathematical demonstration that IS deviation amplifies the bias introduced by PPO clipping, which we formally establish in our work.** Furthermore, our focus is on reinforcement learning in massively parallel environments, which differs from the setting considered by Xie et al. (2025). We therefore believe that the contribution of our paper remains distinct and intact. That said, we truly appreciate the insightful suggestion and support.
> >
> > Thank you again for drawing our attention to this relevant and important work.

---

### Author Response · Authors · 2025-12-03
**Author Final Remarks**

We thank all reviewers and the original area chair for their careful reading and constructive feedback, and we appreciate the newly assigned area chair for their time following the OpenReview incident.

**We summarize the contributions and the additional insights gained during the rebuttal and discussion process.**

# Our Contributions:
**In this paper, we**
- **Theoretically revealed that excessive misalignment between the leader and follower policies** in ensemble policy gradient methods **degrades training stability and sample efficiency**.

- **Introduced a theoretically grounded KL divergence constraint** to encourage followers to explore within a KL-bounded neighborhood around the leader, and **an auxiliary adversarial reward** to prevent overconcentration among followers.

- **Demonstrated that CPO outperforms strong baselines** (including SAPG and DexPBT) across multiple dexterous manipulation tasks in both performance and stability, and **empirically validated that KL constraints reduce density ratio deviation and improve sample efficiency**.

# Reviewers Comments:
Reviewers **highly evaluated our thorough theoretical analysis of ensemble learning dynamics and the proposed method with theoretical justification**, and provided positive scores and comments, especially noting the **significant performance gains on challenging tasks** such as Two-Arms Reorientation. They also praised the **empirical evidence supporting the relationship between KL constraints and effective sample size**, including the **visualizations of KL divergence**.

# During Rebbuttal Process:
We addressed the improvement points raised by the reviewers, especially those from Reviewer FSNQ and Reviewer ZcZ8 as follows:

- ***Is the computational overhead introduced by CPO worthwhile?*** (Reviewer FSNQ \& ZcZ8)

    **Response:** We clarified that although CPO introduces extra backprop passes (12 vs. 7 per iteration, resulting in **\~25% overhead including simulation time**), this cost is outweighed by substantial sample efficiency: **CPO requires over 2× fewer samples** to reach SAPG’s final performance and often **achieves higher final performance** on tasks with strong misalignment (See response to Reviewer ZcZ8).

- ***Does CPO generalize to non-dexterous manipulation tasks?*** (Reviewer FSNQ \& ZcZ8)

    **Response:** We **conducted additional experiments on non-dextrous manipulation tasks**, and confirmed that **CPO outperforms all baselines** in final performance, demonstrating clear generalizability beyond dexterous environments (new results in Fig. 5, Apx. A.5).

-  ***When does misalignment occur between leader and follower policies?*** (Reviewer FSNQ)

    **Response:** We **conducted additional experiments and revealed that strong entropy regularization in SAPG consistently increases inter-policy KL divergence**, causing severe misalignment and destabilizing leader updates. In contrast, CPO suppresses misalignment while still enabling effective exploration. (See Sec. A.8)

- ***Why does a 40× ESS improvement result in only 2–2.5× sample-efficiency gains?*** (Reviewer FSNQ)

    **Response:** We clarified that ESS measures how evenly samples contribute to learning under importance sampling and is related to sample efficiency, but its relationship with expected return is not linear; thus, large ESS improvements mainly reduce gradient variance rather than proportionally accelerating learning.

- ***How sensitive is CPO to hyperparameters?*** (Reviewer ZcZ8)

    **Response:** We clarified that CPO is robust across a wide range of KL-constraint strengths and that the KL constraint is the primary driver of performance gains, with the adversarial reward adding a small but consistent benefit. We also provided practical guidance for tuning the KL coefficient based on inter-policy KL and ESS.

- ***Contribution of the paper*** (Reviewer ZcZ8)

    **Response:** We clarified that **our theoretical findings are significant and help advance ensemble RL** in massively parallel environments, and that despite its simplicity, **our theoretically grounded method** consistently delivers substantial performance improvements.

Based on these discussions, **Reviewer ZcZ8 raised their score from 4 to 6** (before the score rollback), and **two other reviewers (hCse and nLNK) maintained strong acceptance stances with scores of 8**. While Reviewer FSNQ had not responded to our rebuttal, we believe that their concerns have also been fully resolved, as reflected in our responses.

---
We sincerely appreciate all reviewers again for the thoughtful discussion. We believe that we have addressed all raised points and further improved the manuscript.

By combining theoretical analysis and empirical evidence, our work **highlights that appropriately constraining diversity is as important as promoting it in ensemble learning**. We hope these findings will contribute to the future advances in large-scale RL.

---

### Meta-Review · Area_Chair_eHZb · 2026-01-05

**Summary:**

The reviewers broadly agree that the paper presents a well-motivated and technically sound method (Coupled Policy Optimization, CPO) for regulating policy diversity in large-scale ensemble reinforcement learning. The core idea is controlling inter-policy divergence via KL constraints grounded in off-policy stability analysis, and simultaneously incorporating an intrinsic adversarial reward to encourage sufficient separation among the policies. The idea is widely viewed as correct, theoretically justified, and empirically effective, especially on challenging dexterous manipulation tasks.

Concerns primarily focus on incremental novelty, computational overhead, limited task diversity, and the marginal contribution of the adversarial reward. Most of these concerns were substantially addressed in the rebuttal, leading several reviewers to maintain or increase their scores.

Thus, I'm recommending to accept this paper.

**Reviewer Concerns:**

Concerns largely addressed by the rebuttal:
* Rationale and role of the adversarial (DIAYN-based) reward
* Generalizability beyond dexterous manipulation
* Choice of KL divergence and its theoretical justification
* Hyperparameter (lambda) selection guidance
* Source and conditions of policy misalignment

Concerns partially addressed or still outstanding:
* Incremental nature of the algorithmic contribution: While the rebuttal emphasized the theoretical insights and empirical alignment, the core mechanism (KL regularization in a leader–follower setup) remains conceptually simple. This concern is acknowledged but not fully resolvable, as it reflects a judgment about novelty rather than correctness.
* Compute and wall-clock efficiency under controlled conditions: The author's rebuttal provided complexity analysis and time-to-performance arguments.
* Limited breadth of environments and difficulty axes: Although additional tasks were added, experiments still focus on manipulation and locomotion under large-scale parallel settings. Broader validation remains an open direction.

**Reviewer Scores:**

Reviewer ZcZ8 raised their score from 4 to 6.

Reviewer hCse and nLNK maintained their scores as 8.

Reviewer FSNQ would have maintained their score as 4 since their concerns have been partially addressed, as is discussed in Reviewer Concerns part.

---

### Decision · Program_Chairs · 2026-01-26

Accept (Poster)